# Effects of fluorescent glutamate indicators on neurotransmitter diffusion and uptake

**Moritz Armbruster[1], Chris G Dulla[1], Jeffrey S Diamond[2]\***

[1]Department of Neuroscience, Tufts University School of Medicine, Boston, United States; [2]Synaptic Physiology Section, NINDS Intramural Research Program, National Institutes of Health, Bethesda, United States

**Abstract** Genetically encoded fluorescent glutamate indicators (iGluSnFRs) enable neurotransmitter release and diffusion to be visualized in intact tissue. Synaptic iGluSnFR signal time courses vary widely depending on experimental conditions, often lasting 10–100 times longer than the extracellular lifetime of synaptically released glutamate estimated with uptake measurements. iGluSnFR signals typically also decay much more slowly than the unbinding kinetics of the indicator. To resolve these discrepancies, here we have modeled synaptic glutamate diffusion, uptake and iGluSnFR activation to identify factors influencing iGluSnFR signal waveforms. Simulations suggested that iGluSnFR competes with transporters to bind synaptically released glutamate, delaying glutamate uptake. Accordingly, synaptic transporter currents recorded from iGluSnFR-expressing astrocytes in mouse cortex were slower than those in control astrocytes. Simulations also suggested that iGluSnFR reduces free glutamate levels in extrasynaptic spaces, likely limiting extrasynaptic receptor activation. iGluSnFR and lower affinity variants, nonetheless, provide linear indications of vesicle release, underscoring their value for optical quantal analysis.

**\*For correspondence:**
diamondj@ninds.nih.gov

**Competing interests:** The authors declare that no competing interests exist.

## Introduction

Periplasmic binding proteins (PBPs) have been modified to develop genetically encoded biosensors to detect different molecules, including glutamate (*de Lorimier et al., 2002*). PBPs comprise two domains linked by flexible 'hinge' where ligand binding brings the two domains closer together (*Quiocho et al., 1997*). PBP glutamate indicators are based on GltI, part of *E. Coli.*'s ABC glutamate/aspartate transporter complex; early versions (FLIPE and GluSnFR; *Okumoto et al., 2005*; *Hires et al., 2008*) signaled ligand binding via Förster Resonance Energy Transfer (FRET; *Fehr et al., 2002*) between fluorescent proteins tethered to each PBP domain. Limitations due to low FRET efficiency subsequently were overcome with iGluSnFR, a single-fluorophore sensor with circularly permuted GFP inserted near GltI's hinge region so that glutamate binding increases GFP fluorescence (*Marvin et al., 2013*). iGluSnFR variants exhibiting faster glutamate dissociation rates recently have been developed to image glutamate with higher temporal resolution (*Helassa et al., 2018*; *Marvin et al., 2018*). iGluSnFR has been used to detect relative amounts of glutamate release evoked by different physiological stimuli (*Borghuis et al., 2013*; *Yonehara et al., 2013*; *Armbruster et al., 2016*; *Franke et al., 2017*; *Pinky et al., 2018*) and to compare the diffusion lifetime of synaptically released glutamate in different brain regions (*Pinky et al., 2018*). Analogous information has been obtained using synaptically-evoked excitatory amino acid transporter (EAAT)-mediated currents (STCs) recorded in astrocytes (*Bergles and Jahr, 1997*; *Diamond et al., 1998*; *Lüscher et al., 1998*; *Diamond and Jahr, 2000*; *Diamond, 2005*; *Hanson et al., 2015*). Although these two approaches can lead to similar conclusions (e.g. *Hanson et al., 2015*; *Armbruster et al., 2016*; *Pinky et al., 2018*), they differ significantly in their response time courses: Synaptically evoked

iGluSnFR-mediated fluorescence signals decay with exponential time courses ranging from ~20 ms (*Marvin et al., 2013*; *Armbruster et al., 2016*) to 100 ms or more (*Parsons et al., 2016*; *Pinky et al., 2018*). STCs decay nearly 10 times more rapidly (*Bergles and Jahr, 1997*; *Diamond and Jahr, 2000*; *Diamond, 2005*) and suggest that glutamate is removed from the extracellular space by EAATs in just a few ms (*Diamond, 2005*). This difference between iGluSnFR response and STC waveforms is evident even in studies employing both techniques under apparently similar experimental conditions (*Armbruster et al., 2016*).

For iGluSnFR signals and STCs to provide quantitative insights into the dynamics of glutamatergic transmission, the kinetic discrepancies between the two signals must be understood. The STC time course reflects a combination of release asynchrony, transporter kinetics, glutamate diffusion and electrotonic distortion by astrocytic membranes (*Bergles and Jahr, 1997*; *Diamond, 2005*). By contrast, the factors determining iGluSnFR response waveforms have not been identified explicitly. Slower iGluSnFR responses do not simply reflect the kinetics of the indicator, as the iGluSnFR dissociation time course is 2–10-fold faster than most response decays. Moreover, iGluSnFR signals are slowed by partial blockade of glutamate transporters (*Armbruster et al., 2016*; *Parsons et al., 2016*; *Pinky et al., 2018*), indicating that they report changes in uptake capacity and glutamate clearance. Whereas STCs reflect the naturally occurring process of glutamate uptake by endogenous transporters, iGluSnFR expression introduces exogenous binding sites into the extracellular milieu. The extent to which glutamate buffering by iGluSnFR may influence glutamate diffusion is not intuitively obvious.

Here, Monte Carlo simulations of glutamate diffusion, uptake and iGluSnFR signaling were performed to explore the mechanisms underlying iGluSnFR signal dynamics. These simulations show that iGluSnFR response time course depends strongly on iGluSnFR expression level. Simulated iGluSnFR responses mimic those reported in the experimental literature only when iGluSnFRs compete with EAATs for glutamate to the extent that iGluSnFR delays glutamate uptake. These predictions were confirmed with electrophysiological recordings from iGluSnFR-expressing astrocytes in cortical slices: STCs recorded in iGluSnFR-expressing astrocytes rose and decayed more slowly than those recorded in control astrocytes expressing tdTomato, indicating that iGluSnFR expression slowed the glutamate uptake time course. We conclude that, although iGluSnFR and STCs provide powerful, complementary indications of glutamate release and clearance, care is required in their interpretation. Our simulations suggest that an ideal glutamate indicator would exhibit a large dynamic range (i.e. $\Delta F/F_0$) and low expression levels to deliver detectable signals while minimally disrupting glutamate uptake.

## Results

The stochastic behavior of simulated glutamate transporter and iGluSnFR molecules was governed by experimentally derived kinetic models (*Figure 1*). Equilibrium kinetics of simulated EAAT2 (*Bergles et al., 2002*), iGluSnFR and two iGlu variants (iGlu$_f$ and iGlu$_u$; *Helassa et al., 2018*) were examined first by constructing Markov models of each (*Figure 1—figure supplement 1*) and challenging them with 20 ms applications of glutamate at different concentrations (e.g. *Figure 1A*). This approach yielded equilibrium dose-response curves (*Figure 1B*) and affinities ($K_D$; *Figure 1C*, *left*) that matched closely those reported previously (*Bergles et al., 2002*; *Helassa et al., 2018*). iGlu molecules exhibited similar activation kinetics but a range of deactivation (unbinding) kinetics that varied inversely with affinity (*Figure 1C,D*). As the brightness of resting and activated indicator ($F_{off}$ and $F_{on}$, respectively) has been measured (*Helassa et al., 2018*; *Figure 1—figure supplement 1B*), iGlu responses also could be expressed in terms of $\Delta F/F_0$ (i.e., $(F_{on}-F_{off})/F_{off}$; *Figure 1E*).

### Stochastic model of glutamate diffusion, uptake and iGluSnFR activation

Monte Carlo simulations of glutamate release from a single synapse (*Diamond, 2005*; see Materials and methods) comprised pre- and postsynaptic hemispherical compartments (320 nm diameter; *Schikorski and Stevens, 1997*; *Ventura and Harris, 1999*) separated by a 20 nm cleft and surrounded by a three-dimensional, isotropic, abstract representation of extracellular space (*Rusakov and Kullmann, 1998*) populated with EAAT and iGluSnFR molecules at specified concentrations (*Figure 1F*). EAAT1 and EAAT2 are expressed in astrocytic membranes at high density

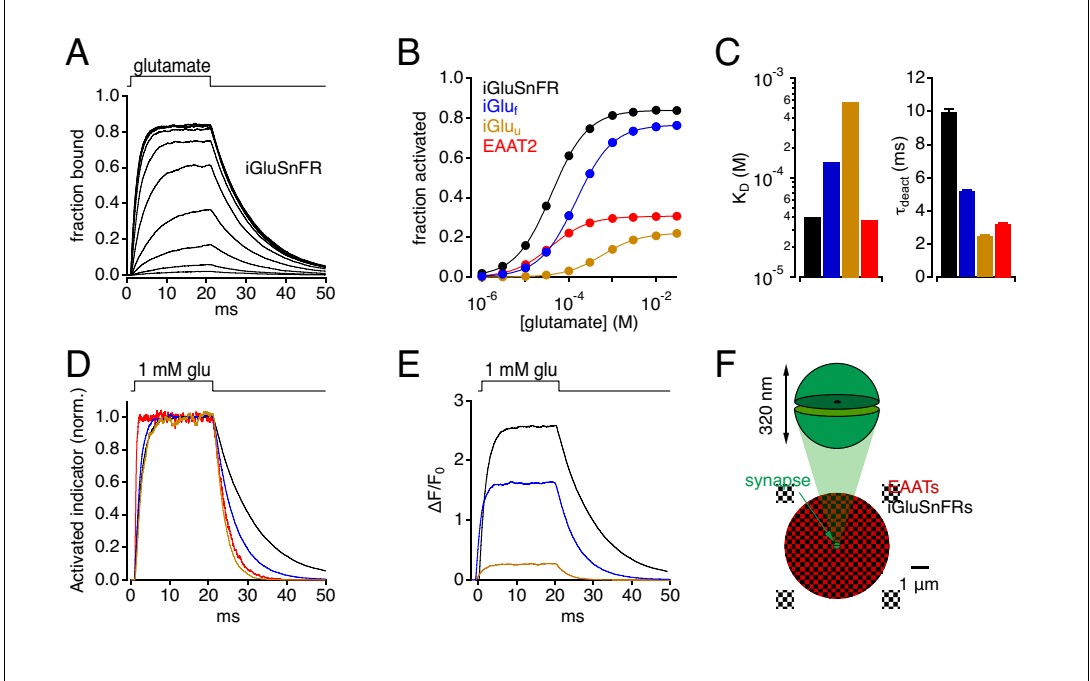

**Figure 1.** Simulations: Kinetic properties of glutamate indicators/transporters and simulated synaptic responses. (**A**) Simulated iGluSnFR activation by 20 ms applications of glutamate (concentration steps varied logarithmically from 1 μM to 30 mM). (**B**) Comparison of simulated glutamate dose-response curves for iGluSnFR (black), iGlu$_f$ (blue), iGlu$_u$ (gold) and EAAT2 (red). Color scheme applies to the entire figure. (**C**) Simulated equilibrium affinities ($K_D$, left) and deactivation time constants (mean ± SD, n = 10 different [glu] applications, right) for iGlus and EAAT2. (**D**) Activation of iGlus and EAAT2 by 1 mM glutamate, normalized and superimposed to compare activation and deactivation kinetics. (**E**) Responses of iGlus to 1 mM glutamate, scaled according to their background fluorescence and change in fluorescence upon activation (i.e., $\Delta F/F_0 \asymp (F_{on}\text{-}F_{off})/F_{off}$; see **Figure 1—figure supplement 1**; **Helassa et al., 2018**). (**F**) Schematic diagrams of simulated synaptic structure (*top*) and surrounding extracellular space (*bottom*). The online version of this article includes the following source data and figure supplement(s) for figure 1:

**Source data 1.** Characterization of kinetic models.
**Figure supplement 1.** Kinetic models.

(>$10^4$ molecules per μm$^2$) that, adjusted for membrane density and extracellular volume fraction, corresponds to an effective concentration of 140–330 μM in the extracellular space of hippocampal and cerebellar neuropil (**Lehre and Danbolt, 1998**). Accordingly, the time course of glutamate uptake in adult CA1 hippocampal astrocytes is well modeled with an active EAAT concentration of about 100 μM (**Diamond, 2005**), the value used here. iGluSnFR concentrations have not been measured but, because they may vary widely depending on factors influencing expression, we simulated a large range (1–3000 μM).

5000 glutamate molecules were released at the center of the synaptic cleft, with each individual glutamate molecule undertaking a random walk slowed by the tortuosity of the extracellular space (**Nicholson and Syková, 1998**; **Rusakov and Kullmann, 1998**; **Nielsen et al., 2004**; **Diamond, 2005**). The extrasynaptic space was populated evenly with completely overlapping distributions of EAATs and iGluSnFRs. Because glial and neuronal membranes are so closely apposed in synaptic neuropil (**Mishchenko et al., 2010**), the large majority of glial EAATs likely must compete with iGluSnFRs for glutamate, regardless of whether iGluSnFR is expressed in neurons or in glia (**Armbruster et al., 2016**). Additional experiments, presented below (Figure 9), examined other extrasynaptic expression patterns of EAATs and iGluSnFRs.

## Competition between iGluSnFRs and EAATs slows uptake and iGluSnFR activation time courses

When EAATs (100 μM) and iGluSnFRs (300 μM) were co-localized in extrasynaptic space, simulated glutamate release activated iGluSnFR with a time course that reached a peak in about 5 ms and decayed with an exponential time course (τ = 27 ms; **Figure 2A**). This decay was significantly slower

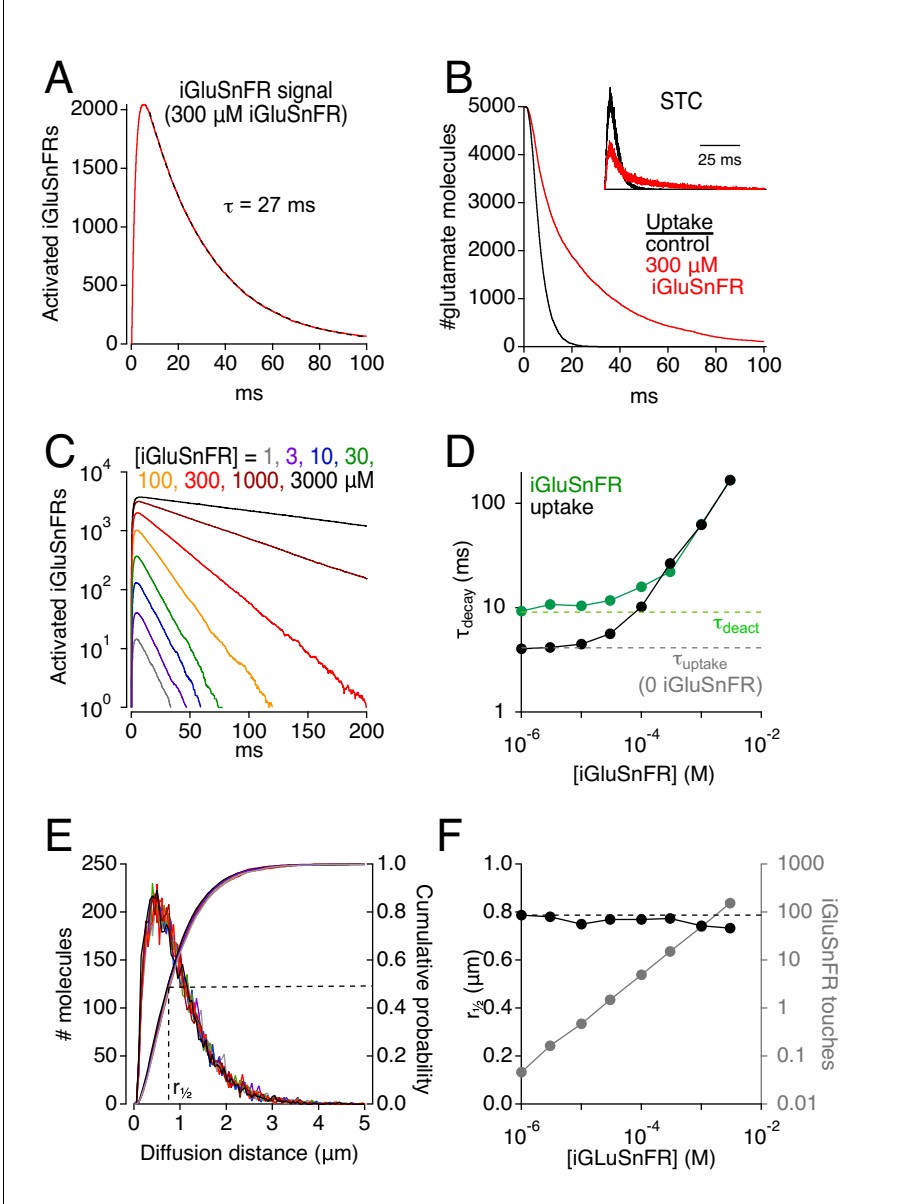

**Figure 2.** Simulations: iGluSnFR can distort fluorescence and glutamate clearance time courses. (**A**) Simulated iGluSnFR activation elicited by the release of 5000 glutamate molecules from the center of the cleft in the scheme depicted in *Figure 1F*. Dashed line indicates exponential fit to the response decay. Activation measured over a region of interest (ROI, radius = 10 μm) centered about the synapse. (**B**) Simulated time course of glutamate uptake in the absence of iGluSnFR (black) and in the presence of 300 μM iGluSnFR (red). Inset, simulated STCs in the presence (red) and absence (black) of 300 μM iGluSnFR. STCs are typically inward (negative) currents but are inverted here for simplicity. (**C**) The concentration of iGluSnFR influences its activation time course. A semi-log plot shows that the exponential decay slows as [iGluSnFR] increases (ROI radius = 10 μm). (**D**) Summarized data from C (green). Glutamate uptake (black) is also slowed by [iGluSnFR]. (**E**) iGluSnFR buffering does not affect the distance that glutamate diffuses prior to being taken up by EAATs. Same color scheme as in C. Distance measured from the center of the synaptic cleft. (**F**) Summarized data from E. Gray trace shows the average number of times each glutamate molecule binds to iGluSnFR prior to being taken up by an EAAT.

The online version of this article includes the following source data for figure 2:

**Source data 1.** Simulated STCs and iGluSnFR signals.

than iGluSnFR's deactivation time constant ($\tau$ = 10 ms; *Figure 1C*, *right*), suggesting that iGluSnFR interacted with extrasynaptic glutamate over a prolonged period. Consistent with this, the time course of glutamate uptake from the extracellular space was slowed in the presence of iGluSnFR (*Figure 2B*). This slowing was also evident in the simulated STC (*Figure 2B*, inset), which reflects electrogenic state transitions within EAATs upon binding and transporting glutamate (*Bergles et al., 2002*; see Materials and methods). The time courses of both the iGluSnFR response and glutamate uptake were prolonged by higher iGluSnFR concentrations (*Figure 2C,D*), suggesting that iGluSnFR buffers glutamate diffusion and delays its uptake. At very low iGluSnFR levels (e.g. 1 µM), the uptake time course closely approximated the room temperature control in the absence of iGluSnFR ($\tau$ ~4 ms; *Diamond, 2005*), and iGluSnFR activation decayed at a rate approaching iGluSnFR deactivation (*Figure 2D*). At higher iGluSnFR concentrations, however, both time constants increased and converged; the highest iGluSnFR levels tested gave rise to time constants that exceeded 100 ms (*Figure 2D*), similar to slower published iGluSnFR signal time courses (*Parsons et al., 2016*; *Pinky et al., 2018*). Although iGluSnFR extended the extracellular lifetime of glutamate, it did not affect the distance that glutamate diffused prior to being taken up by transporters (*Figure 2E,F*), consistent with iGluSnFR's role as a stationary buffer. At higher expression levels, each glutamate molecule bound to iGluSnFR multiple times before it was able to bind a transporter (*Figure 2F*). (Each glutamate molecule typically bound a transporter only once, as simulated EAATs transported glutamate with 90% efficiency.) These results indicate that strong iGluSnFR buffering slows glutamate uptake.

## iGluSnFR reduces extrasynaptic glutamate concentration and receptor activation

If iGluSnFR buffered synaptically released glutamate, it should reduce the concentration of free neurotransmitter in the extrasynaptic space, thereby decreasing activation of glutamate receptors perisynaptically and, perhaps, in neighboring synapses. To test this, we measured the simulated free (unbound, untransported) glutamate concentration in three regions: 1) in the center of the synaptic cleft, above the PSD ($\leq$110 nm from the release site); 2) in the extrasynaptic region immediately surrounding the cleft (160–260 nm from the release site); and 3) in a more distant extrasynaptic region approximating the average distance between neighboring synapses (400–500 nm from the release site; *Rusakov and Kullmann, 1998*). Glutamate concentration was measured following the release of 5000 neurotransmitter molecules at the center of the cleft, with 100 µM EAATs and 1–3000 µM iGluSnFR expressed extrasynaptically (*Figure 3*). The acquired concentration profiles were then used to challenge Markov state models of ionotropic and metabotropic glutamate receptors (AMPARs, NMDARs and mGluRs; *Jonas et al., 1993*; *Lester et al., 1993*; *Marcaggi et al., 2009*) to estimate activation of each receptor type in each region.

Glutamate concentration in the synaptic cleft declined rapidly following release ($\tau$ ~33 µs; *Figure 3Ai*), due primarily to diffusion down a steep concentration gradient (*Clements, 1996*; *Wahl et al., 1996*; *Diamond and Jahr, 1997*; *Barbour, 2001*). Accordingly, the glutamate concentration and consequent receptor activation within center of the cleft were unaffected by the presence of extrasynaptic iGluSnFR (*Figure 3Ai–v*).

Perisynaptic glutamate concentration waveforms were smaller, reflecting rapid dilution, yet still relatively brief ($\tau$ ~ 125 µs; *Figure 3Bi*). Consequently, perisynaptic receptors of all types were activated with low probability, even in the absence of iGluSnFR (*Figure 3Biii–v*); extrasynaptic iGluSnFR reduced the amplitude and sped the time course of the perisynaptic glutamate concentration, thereby reducing perisynaptic receptor activation (*Figure 3B*). Similar results were observed 400–500 nm from the release site, although receptor activation in this region was extremely low in any condition (*Figure 3C*; see also *Barbour, 2001*). Taken together, these results suggest that the buffering actions of iGluSnFR may reduce primarily activation of perisynaptic receptors, including mGluRs, potentially influencing homosynaptic modulation of synaptic strength (*Bashir et al., 1993*; *Kato, 1993*).

If expressed under control of a neuronal promoter, iGluSnFR might be present in the synaptic cleft as well as extrasynaptic regions. We simulated this scenario next by including iGluSnFR in all extracellular spaces, including the synaptic cleft (*Figure 3D*). Only the highest levels of iGluSnFR-bound glutamate rapidly enough to speed the decay of the free glutamate concentration transient in the cleft (*Figure 3Di*). iGluSnFR's buffering action also slowed glutamate's escape from the cleft,

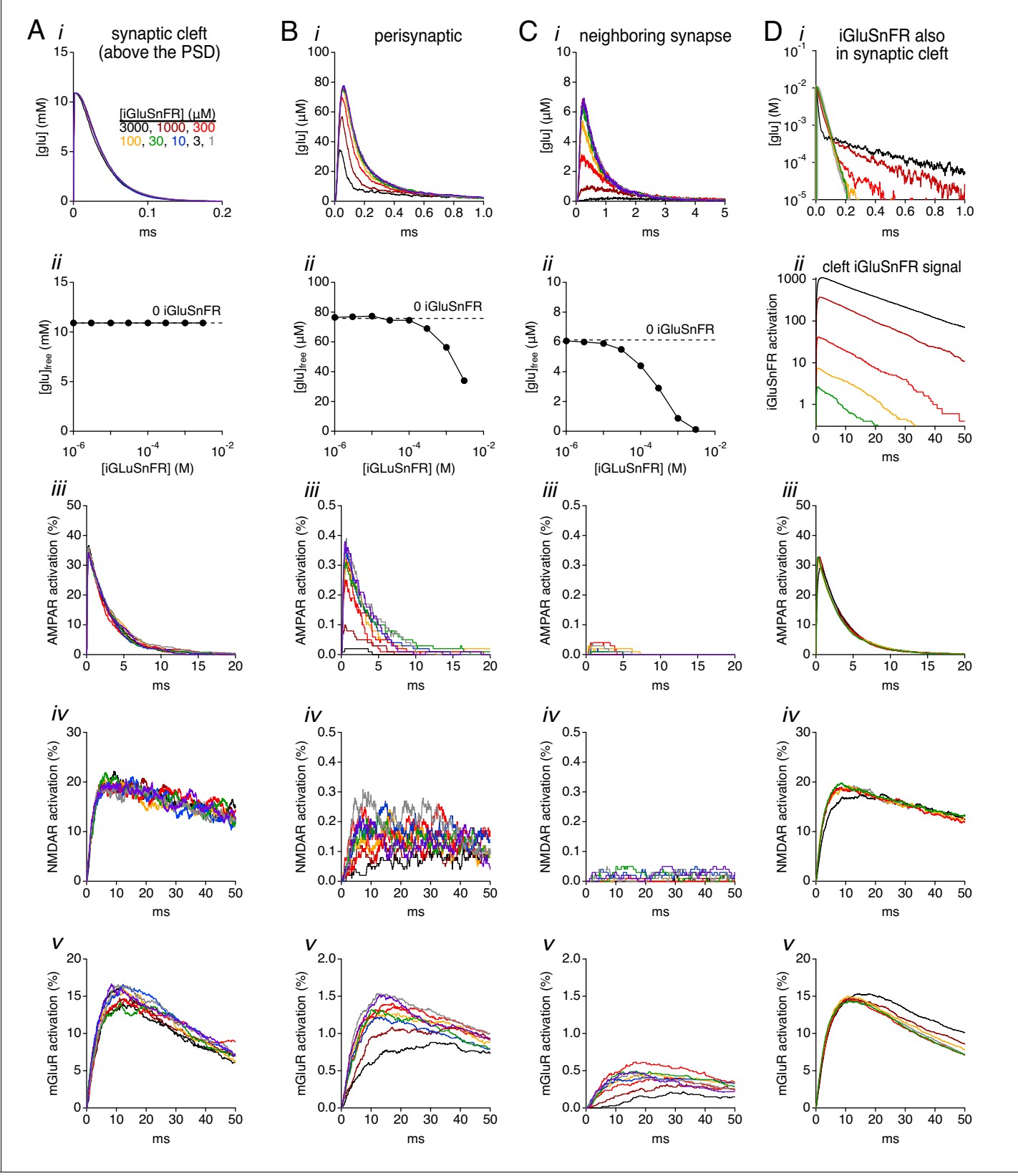

**Figure 3.** Simulations: effects of iGluSnFR on glutamate concentration and receptor activation. (**A**) Simulated glutamate waveforms (i), peak glutamate concentration (ii), AMPAR activation (iii), NMDAR activation (iv) and mGluR activation (v) in the synaptic cleft (≤110 nm from the release site). Trace

*Figure 3 continued on next page*

*Figure 3 continued*

colors correspond to different iGluSnFR concentrations, as indicated in i. (**B**) As in A, but in the perisynaptic region (160–260 nm from the release site). (**C**) As in A, but in the extrasynaptic region 400–500 nm from the release site. (**D**) When iGluSnFR was also present within the synaptic cleft, it influenced cleft glutamate concentration waveforms (i) and produced sizeable signals within the cleft (ii), but did not strongly influence receptor activation within the cleft (iii-v).

The online version of this article includes the following source data and figure supplement(s) for figure 3:

**Source data 1.** Peak free glutamate concentration as a function of iGluSnFR concentration.

**Figure supplement 1.** Simulations: receptor activation and iGluSnFR signals at a lower diffusion coefficient.

however, giving rise to a small, slower component of free glutamate, representing the minor fraction unbound by iGluSnFR (*Figure 3Di*). Together, these changes caused little difference in receptor activation within the cleft (*Figure 3Diii–v*).

Simulated glutamate concentration waveforms were very brief (*Figure 3Ai,Bi,Ci*; see also *Wahl et al., 1996*; *Barbour, 2001*), even though glutamate's diffusion coefficient (D) was set to one-third that measured in aqueous solution (*Longsworth, 1953*) to approximate physiologically estimated values in the synaptic cleft (*Nielsen et al., 2004*). Rapid dilution led to low levels of perisynaptic receptor activation (*Figure 3B*), causing us to wonder how much simulated iGluSnFR signals depend on the chosen value for D. To test this, simulations were repeated with D reduced another fivefold (*Figure 3—figure supplement 1*). Slower diffusion gave rise to longer concentration waveforms and enhanced receptor activation, as expected (*Figure 3—figure supplement 1A–C*), but iGluSnFR exerted similar effects on glutamate concentration and receptor activation; the amplitude and time course of simulated iGluSnFR signals at all indicator concentrations were similar to those acquired at higher D values (*Figure 3—figure supplement 1D,E*).

## iGluSnFR expression slows STCs in cortical astrocytes

The simulations presented so far suggest that iGluSnFR delays glutamate uptake by competing with EAATs and predict that iGluSnFR expression would slow the STC time course. To test this, we recorded STCs in cortical astrocytes from mice expressing either iGluSnFR or tdTomato under control of glial fibrillary acidic protein (GFAP) promoters (GFAP or GfaABC1D, respectively; *Figure 4*). Although GFAP promoters target neural progenitors in addition to astrocytes (*Garcia et al., 2004*), in our hands GFAP-driven expression in adult cortex appears almost exclusively in astrocytes (*Armbruster et al., 2016*). Consistent with the model's predictions, STCs in iGluSnFR$^+$ astrocytes rose and decayed more slowly than those in tdTomato$^+$ astrocytes ($t_{rise}$(10–90%): 6.6 ± 1.8 ms, n = 23 vs. 5.0 ± 2.1 ms, n = 15, $t$(26.1) = 2.42, p=0.023, $t$-test; $\tau_{decay}$: 19.2 ± 5.6 ms, n = 23 vs. 16.0 ± 3.6 ms, n = 15, $t$(36.0) = 2.12, p=0.04, $t$-test; *Figure 4A–D*). This was not due to any apparent changes in astrocyte intrinsic electrical properties: iGluSnFR$^+$ and tdTomato$^+$ astrocytes exhibited similar input resistance (iGluSnFR: 2.9 ± 2.6 MΩ; n = 22; tdTomato: 3.6 ± 2.6 MΩ, n = 16; $t$(32)=-0.75, p=0.46, $t$-test), and iGluSnFR$^+$ astrocytes actually exhibited slightly more hyperpolarized resting membrane potentials (RMP; −73.3 ± 3.0 mV, n = 22 vs. −70.0 ± 4.0 mV, n = 15; $t$(24)=-2.73, p=0.011, $t$-test; *Figure 4E*) that, if anything, would increase slightly the efficacy of glutamate uptake (*Wadiche et al., 1995*). RMP changes were unlikely due to effects on uptake capacity or EAAT expression, as differences in RMP were not observed between WT astrocytes and those recorded in EAAT1$^{+/-}$ or EAAT2$^{+/-}$ mice (J. Shih and C. Dulla, personal communication). These changes in STC waveform occurred under experimental conditions that yielded iGluSnFR decay time courses ($\tau$ = 21.1 ± 4.9 ms, n = 10; *Figure 4F*) that are near the fast end of the range of published iGluSnFR signals and correspond to responses simulated with 300 μM iGluSnFR (*Figure 2D*).

Next, we increased iGluSnFR expression by targeting iGluSnFr to both neurons (hSyn-iGluSnFr) and astrocytes (GFAP-iGluSnFr) and compared STCs and iGluSnFR signals to those recorded in mice doubly infected with hSyn-EGFP and GfaABC1D-tdTomato. As predicted by the simulations, STCs recorded in astrocytes from doubly iGluSnFR-infected mice were slowed to a greater extent compared to control than those in singly-infected mice (*Figure 4G*). Similar results were observed in the iGluSnFR signals (*Figure 4H*; see Materials and methods). Astrocytes from doubly infected iGluSnFr mice exhibited similar membrane properties to those in doubly infected controls, with no significant differences in RMP.

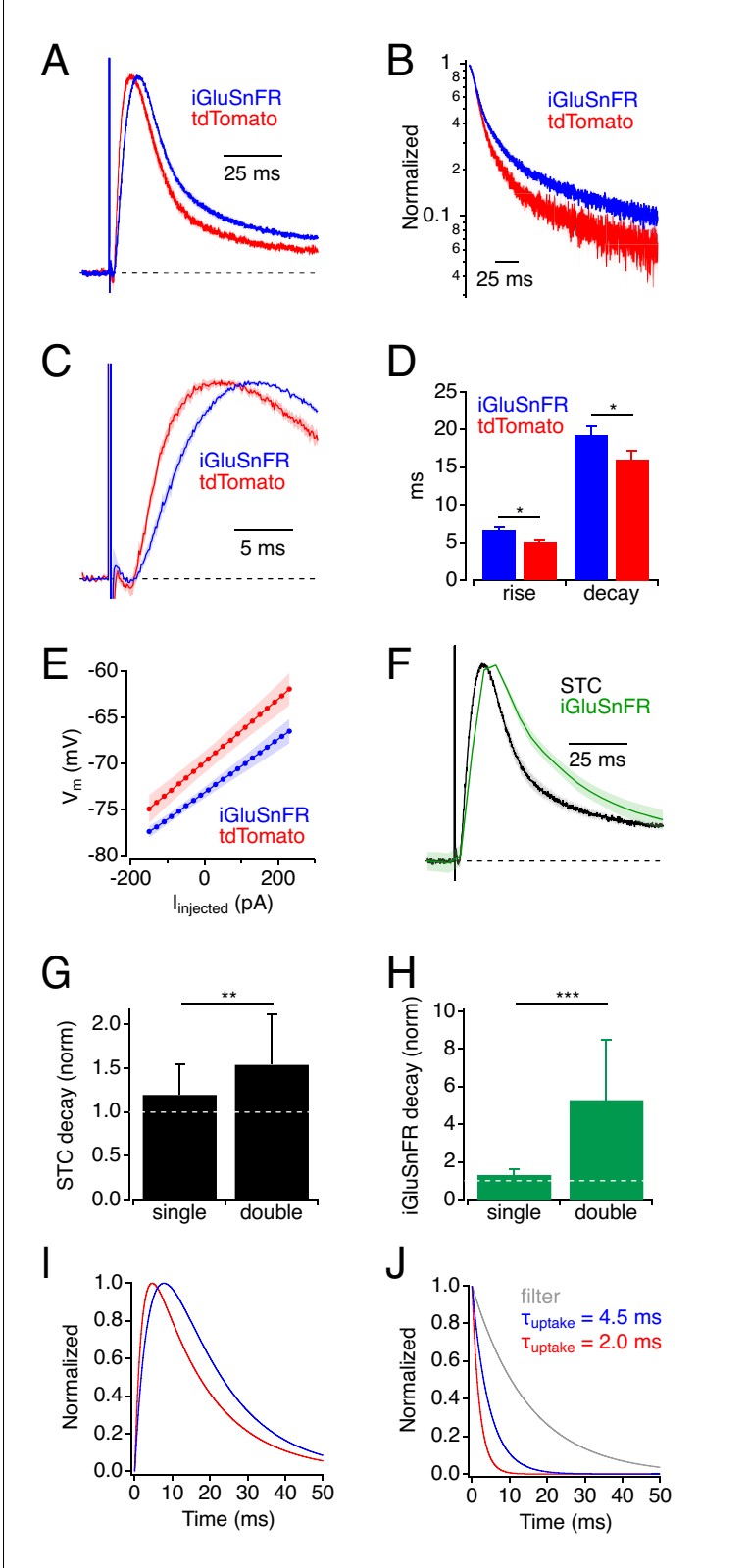

**Figure 4.** Physiological experiments: iGluSnFR expression slows uptake by cortical astrocytes. (**A**) Synaptic transporter currents (STCs) recorded in cortical astrocytes expressing either iGluSnFR (blue) or tdTomato (red). Traces indicate average responses (mean ± SEM; iGluSnFR: n = 21 cells; tdTomato: n = 15 cells), normalized in amplitude. (**B**) Responses in A, plotted on a semi-log scale. (**C**) Rising phases (± SEM) of the responses in A,

*Figure 4 continued on next page*

*Figure 4 continued*

plotted on an expanded time scale. (D) Summary data showing that STC rise and decay were slower in iGluSnFR$^+$ astrocytes. * indicates p<0.05. (E) Astrocyte $V_m$ (mean ± SEM), as a function of injected current, shows that iGluSnFR$^+$ astrocytes rested at slightly more hyperpolarized potentials compared to tdTomato$^+$ astrocytes. Input resistances (indicated by the slope of the relation) were not different in the two groups, although intercell variability diminished the statistical power of this comparison (power = 0.8 would require n = 250). (F) STCs and iGluSnFR signals measured in the same experiments (mean ± SEM, n = 10 cells). (G) Expressing iGluSnFR in both neurons and glia ('double') slowed STCs to a greater extent than when iGluSnFR was expressed in astrocytes only ('single'). Asterisks indicate p=0.007 (Wilcoxon Rank test between single (n = 15) and double (n = 13)). Decay time constants normalized to average STC decay in control. (H) As in G, but showing the decays of iGluSnFR signals. Asterisks indicate p=0.0003 (Wilcoxon Rank test between single (n = 10) and double (n = 5)). (I) Simulated STC waveforms corresponding to average responses in iGluSnFR$^+$ (blue) and tdTomato$^+$ (red) astrocytes from panel A. (J) Waveforms used to derive STCs in I. In each case a clearance time course (red or blue) was convolved with a filter waveform (gray). This simple example demonstrates how even subtle differences in STC time course can reflect substantial differences in glutamate clearance time course (*Diamond, 2005*).

The online version of this article includes the following source data for figure 4:

**Source data 1.** STC and iGluSnFR imaging experiments.

At first glance, the observed differences in STC time course (e.g. *Figure 4A*) might appear relatively subtle, but they likely indicate substantial modifications of the glutamate clearance time course. Previous analyses of STCs in CA1 hippocampal astrocytes showed that the STC waveform reflects the time course of glutamate uptake filtered primarily by the electronic properties of the astrocyte (*Diamond, 2005*). This filtering slows both the rise and decay of the STC and obscures the actual time course of glutamate clearance. Consequently, even small changes in STC waveform likely indicate significant changes in glutamate clearance (e.g. *Figure 4G,H*). Note that simulated STCs and derived uptake times do not reflect any electrotonic distortion or release asynchrony.

## iGluSnFR signal time course and ΔF/F$_0$ depends on imaging volume

iGluSnFR enables glutamate to be imaged over a range of spatial scales, from a < 1 μm synapse to an entire brain region (*Marvin et al., 2013*). Glutamate clearance from a synapse is driven primarily by diffusion down a locally steep concentration gradient (*Wahl et al., 1996*; *Diamond and Jahr, 1997*; *Barbour, 2001*), so that the fractional reduction in glutamate concentration is fastest close to the point of release (*Barbour and Häusser, 1997*). Accordingly, simulations indicated that iGluSnFR activation signals were faster when measured over a smaller spherical region of interest (ROI) surrounding the release site (*Figure 5A,B*), consistent with experimental results (e.g. *Marvin et al., 2013*). This volume-dependent effect was greater at higher iGluSnFR concentrations because stronger buffering prolonged further the extrasynaptic lifetime of glutamate (*Figure 5B*). At higher expression levels, iGluSnFR bound simultaneously a significant fraction of synaptically released glutamate, approaching levels limited by the kinetic maximum probability of fluorescence ($P_{max}$, analogous to the maximal open probability of an ion channel; *Figures 5C* and *1B*). Consequently, the iGluSnFR activation coefficient of variability (CV = σ/mean) decreased at higher iGluSnFR concentrations (*Figure 5D*).

The brightness of iGluSnFR has been measured in the unbound and activated states (e.g. *Helassa et al., 2018*), allowing simulated iGluSnFR activation to be expressed in terms of fluorescence, typically reported as the change in fluorescence relative to resting levels (ΔF/F$_0$; *Figure 5E*). Because inactive iGluSnFR in extrasynaptic tissue contributes to F$_0$, the single synapse ΔF/F$_0$ decreased dramatically over larger imaging volumes (*Figure 5E,F*), underscoring the necessity of highly localized laser scanning fluorescence microscopy for examining iGluSnFR signals at single synapses (*Helassa et al., 2018*; *Marvin et al., 2018*). When compound iGluSnFR responses are recorded from many synapses simultaneously, often using wide-field imaging techniques (e.g. *Figure 4*), both the number of imaged synapses and the background fluorescence increase proportionally to imaging volume, so that ΔF/F$_0$ does not vary greatly over most ROI dimensions (*Figure 5G*). Note that ΔF/F$_0$ values decreased as iGluSnFR expression level increased (*Figure 5F,G*). This effect appears due primarily to the increasing F$_0$, not a significant decrease in the fraction of iGluSnFR bound, because even at very low iGluSnFR concentrations glutamate bound only a small fraction of

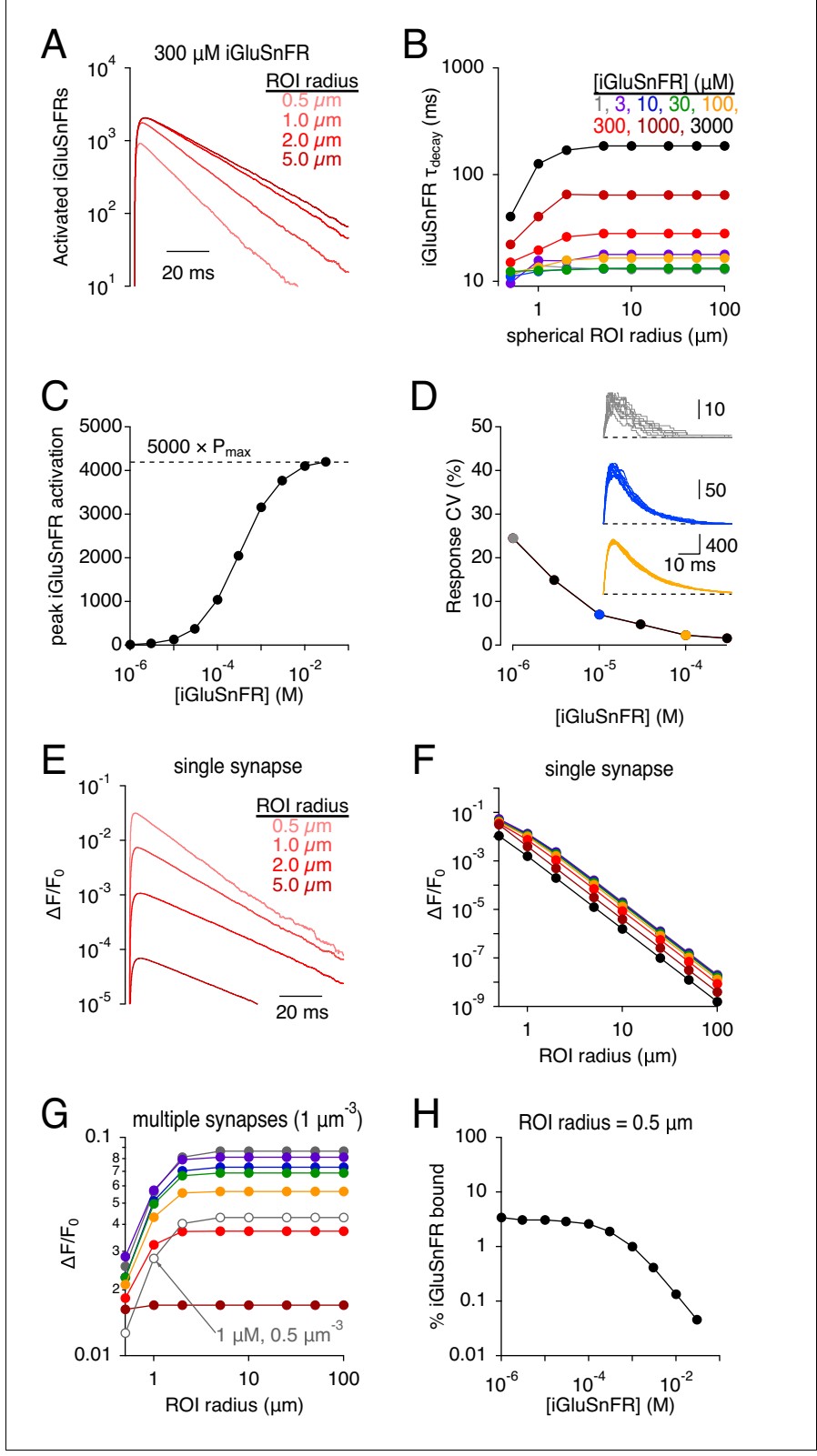

**Figure 5.** Simulations: iGluSnFR signal time course and SNR depends on the imaging volume. (**A**) iGluSnFR (300 µM) signals measured across different spherical regions of interest (ROIs). (**B**) Summary data shows that the dependence on ROI volume is greater at higher iGluSnFR concentrations. (**C**) Peak iGluSnFR response to the release of 5000 glutamate molecules as a function of iGluSnFR concentration. Dashed line indicates maximal

*Figure 5 continued on next page*

*Figure 5 continued*

signal based on maximal occupancy (see *Figure 1B*). (D) iGluSnFR signal variability decreases with indicator concentration. Inset, individual responses at three different iGluSnFR concentrations (gray, 1 µM; blue, 10 µM; orange, 100 µM). (E-F) iGluSnFR single-synapse response signal ($\Delta F/F_0$) depends on indicator concentration (E) and ROI dimensions (F). (G) iGluSnFR compound $\Delta F/F_0$ responses depend on ROI dimensions and the density of activated synapses. (H) When iGluSnFR was evenly sampled throughout even a small volume, only a small fraction of the indicator was activated by glutamate. Laser line scanning, by contrast, yields higher $\Delta F/F_0$ values (*Helassa et al., 2018*).

The online version of this article includes the following source data for figure 5:

**Source data 1.** iGluSnFR signals depend on imaging volume.

iGluSnFR within a 0.5 µm-radius sphere surrounding the synapse (*Figure 5H*). These results point to a counterintuitive conclusion that increased iGluSnFR expression may, in many experimental conditions, actually decrease $\Delta F/F_0$ values of synaptic responses.

## Effects of blocking EAATs on iGluSnFR signals

The rate at which glutamate is taken up from the extracellular space depends critically on EAAT expression levels in astroglia (*Bergles and Jahr, 1997*; *Diamond and Jahr, 2000*; *Diamond, 2005*; *Thomas et al., 2011*). In the hippocampus, the EAAT2 subtype constitutes ~ 80% of glial glutamate transporters, and the remaining 20% are EAAT1 (*Lehre and Danbolt, 1998*). A recent study reported large differences between the effects of an EAAT2-selective antagonist and complete EAAT blockade by a pan-EAAT antagonist on iGluSnFR response time course in hippocampus, leading the authors to suggest that EAAT1 may play a particularly large role in glutamate uptake (*Pinky et al., 2018*). By contrast, STC recordings suggest that blocking EAAT2 slows glutamate uptake by about five-fold (*Diamond and Jahr, 2000*; *Diamond, 2005*), consistent with the relative expression levels of the two transporter subtypes (*Lehre and Danbolt, 1998*). To examine this discrepancy, we simulated the effects of changing transporter density on iGluSnFR activation time course (*Figure 6*). The localization and kinetic properties of EAATs remained the same: only transporter density was changed. The apparent effects of simulated EAAT blockade depended dramatically on the imaging volume. Over a 1 µm radius ROI, the effects on iGluSnFR time course of removing 80% or 100% of the transporters was relatively minor (*Figure 6A*), because the glutamate concentration time course over that small spatial scale is dominated by diffusion, not glutamate uptake (*Diamond and Jahr, 1997*). By contrast, iGluSnFR responses measured over a 5 µm-radius ROI were slowed dramatically by reducing EAAT density, with a particularly large difference observed between 80% and 100% blockade (*Figure 6B*). Note that a 5-µm-radius ROI reported accurately the (iGluSnFR-buffered) time course of glutamate clearance across a range of EAAT levels (*Figure 6F*). These results suggest that a) using iGluSnFR to evaluate manipulations of glutamate diffusion and uptake requires careful consideration of imaging parameters, and b) even a small fraction of expressed EAATs clears glutamate relatively quickly, such that reducing EAATs from control to 20% exerts less dramatic effects on iGluSnFR signals than reducing EAATs from 20% to zero, regardless of EAAT subtype (*Figure 6B*; *Pinky et al., 2018*).

## Lower-affinity iGluSnFR variants do not eliminate buffering artifacts

Recent molecular modifications of iGluSnFR have produced variants that exhibit lower affinity for glutamate, typically by speeding glutamate unbinding (*Figure 1B–D*, *Figure 1—figure supplement 1*; *Helassa et al., 2018*; *Marvin et al., 2018*). These variants may therefore provide the faster fluorescent signals required to image glutamate dynamics accurately, potentially at individual synapses, and resolve individual responses during high-frequency stimulation (*Helassa et al., 2018*; *Marvin et al., 2018*). To test how decreased affinity might influence glutamate signaling, iGluSnFR was replaced in the model by one of two lower affinity variants, iGlu$_u$ or iGlu$_f$, with well-characterized kinetic properties (*Helassa et al., 2018*; *Figure 7*). Both indicators gave rise to faster responses than did iGluSnFR (*Figure 7A,C* and *Figure 2C*) but both produced signal time courses that varied with indicator concentration and imaging volume (*Figure 7A–D*).

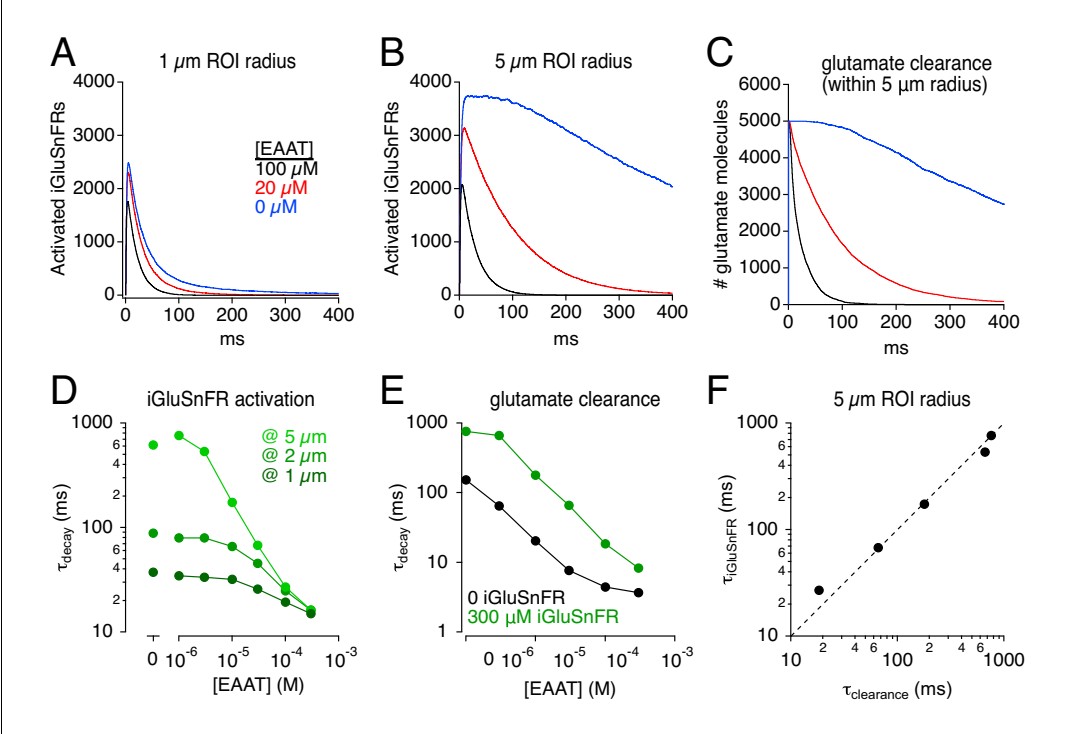

**Figure 6.** Simulations: iGluSnFR distorts effects of varying uptake capacity. (A) iGluSnFR responses (300 µM iGluSnFR) simulated in three different EAAT concentrations. In the immediate vicinity of the synapse (1 µm ROI radius), reducing uptake capacity has little effect on iGluSnFR signal amplitude or time course. (B) As in A, but over a larger imaging volume (5 µm ROI radius), which exaggerates the effects of reducing uptake capacity. (C) Time course of glutamate clearance (uptake and diffusion beyond a 5 µm radius) in the conditions shown in A and B. (D) Summary graph showing the effects of EAAT concentration and ROI dimensions on the exponential decay time course of iGluSnFR activation. (E) The time course of glutamate clearance in the absence of indicator (black) and in the presence of 300 µm iGluSnFR (green). (F) The iGluSnFR signal measured across a 5-µm-radius ROI accurately reports the (modified) time course of glutamate clearance across a range of EAAT concentrations.

The online version of this article includes the following source data for figure 6:

**Source data 1.** Effects of varying [EAAT] on iGluSnFR signals.

## iGluSnFR provides linear indication of glutamate release

iGluSnFR and its variants are potentially valuable tools for comparing relative amounts of glutamate released under different experimental conditions. STCs provide accurately proportionate indications of glutamate release (*Diamond et al., 1998*; *Lüscher et al., 1998*), but similar calibrations of iGluSnFRs have not been performed. To test this, we simulated coincident neurotransmitter release from variable numbers of synapses arranged at different densities (*Figure 8*). The diffusion medium included 100 µM EAAT and 300 µM iGluSnFR, a combination that produced consistent, sizeable iGluSnFR signals at individual synapses (*Figure 2A*) and approximated experimentally observed iGluSnFR time courses (*Figure 4F*). In these multi-synapse simulations, the $30 \times 30 \times 30$ µm$^3$ diffusion space was mapped in Cartesian coordinates and partitioned into $0.1 \times 0.1 \times 0.1$ µm$^3$ transparent cubes. Synapse clusters were arrayed in a 3D hexagonal grid that was centered within the simulation volume and expanded or shrunk to vary spacing between synapses (*Figure 8A,C*; see Materials and methods); to limit any synapse orientation bias, individual synapses were modeled without pre- and postsynaptic processes. Control simulations confirmed that this simplification did not affect iGluSnFR response amplitude or time course when imaged over $\geq$ 2 µm-radius ROIs (*Figure 8—figure supplement 1*).

Excitatory synapses are densely expressed in the CA1 region of the hippocampus, with a 465 nm average distance between nearest neighboring synapses (nearest neighbor distance, NND; *Rusakov and Kullmann, 1998*). When 15 simulated synapses were activated concomitantly at this density, iGluSnFR responses exhibited near perfect linearity, that is the compound response from 15

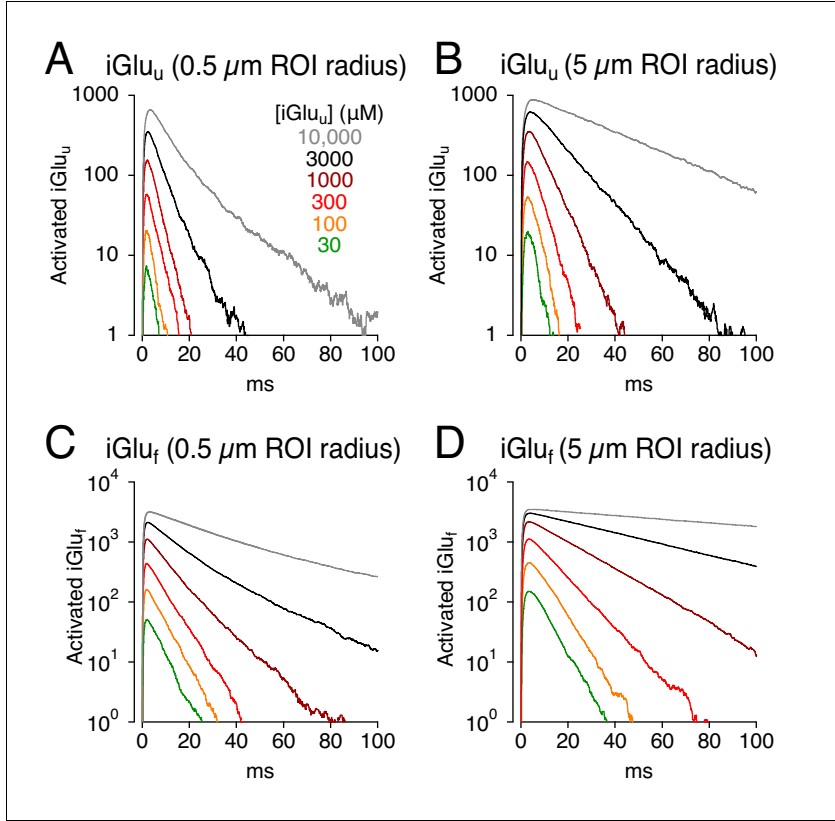

**Figure 7.** Simulations: faster glutamate indicators do not eliminate distortion. (**A**) iGlu$_u$ signals (0.5 µm ROI radius) simulated with different concentrations of the lower-affinity indicator. (**B**) As in A but over a 5 µm ROI radius. (**C and D**), As in A and B but with iGlu$_f$.

synapses was 15.6 times as large as the response from a single synapse (*Figure 8B*). Similar results were observed with iGlu$_f$ and iGlu$_u$ (*Figure 8E–G*). The slight supralinearity likely was due to slightly sublinear uptake reflecting a high degree of EAAT occupancy between activated synapses (data not shown).

Due to their extremely dense expression in astroglial membranes (*Lehre and Danbolt, 1998*), EAATs typically are not saturated even during trains of synaptic stimulation (*Diamond and Jahr, 2000*), enabling STCs to provide a linear indication of glutamate release (*Diamond et al., 1998*; *Lüscher et al., 1998*). This is likely due to the fact that, because the release probability of individual CA1 synapses is ~ 0.3 (*Stevens and Wang, 1995*; *Hjelmstad et al., 1997*), synaptic stimulation is unlikely to evoke coincident release at every synapse. Accordingly, when the NND of activated synapses was increased to 1 µm, simulated uptake rates and iGluSnFR responses exhibited near perfect linearity (*Figure 8D*; uptake data not shown).

Response variability is another way of measuring relative differences in the number of activated synapses (*Faber and Korn, 1991*). Specifically, if individual synapses exhibit similar binomial behavior to each other, the CV of the compound response will vary inversely with the square root of the number of activated synapses. Simulated iGluSnFR responses generally obeyed this relationship, as indicated in a plot of $CV^{-2}$ vs. the number of activated synapses (*Figure 8H*). The slight deviation from the proportional relationship likely reflects subtle differences in quantal variability depending on the physical location of the synapse within the hexagonal array.

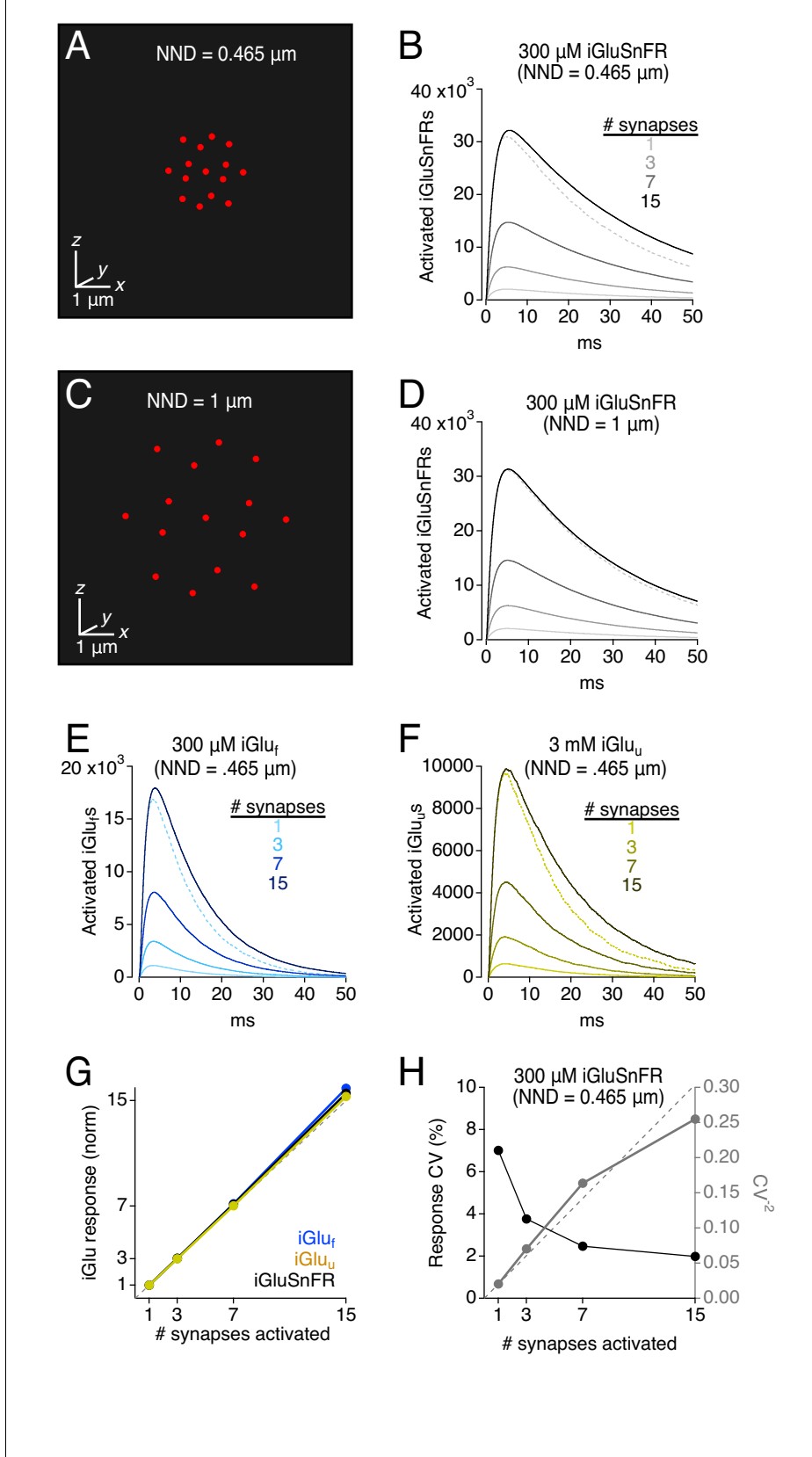

**Figure 8.** Simulations: iGluSnFR and variants provide a linear indication of synaptic release. (**A**) Schematic showing 15 active synapses clustered tightly (NND = 0.465 μm) in 3D diffusion space. (**B**) Simulated iGluSnFR signals (300
*Figure 8 continued on next page*

*Figure 8 continued*

µm iGluSnFR, 100 µm EAAT) elicited by coincident activation of 1, 3, 7, and 15 synapses (NND = 0.465 µm). Dotted gray trace shows the linear prediction for the 15-synapse case, that is the single synapse response multiplied by 15. (**C and D**) As in A and B, but NND = 1 µm. (**E**) As in B but for simulated iGlu$_f$-mediated signals. (**F**) As in B but for simulated iGlu$_u$-mediated signals. (**G**) Summary graph showing that iGlu signals provide a linear indication of synaptic release, even when NND = 0.465. (**H**) iGluSnFR response variability (CV) also provides a reliable indication of relative numbers of activated synapses (see *Faber and Korn, 1991*).

The online version of this article includes the following source data and figure supplement(s) for figure 8:

**Source data 1.** Multi-synapse simulations.

**Figure supplement 1.** Simulations: including the synaptic cleft has little effect on simulated iGluSnFR time courses.

## Segregating EAATs and iGluSnFRs reduces buffering effects but does not speed iGluSnFR signal

It is possible that the buffering effects of iGluSnFR could be ameliorated by segregating EAATs and iGluSnFR into separate compartments, thereby giving EAATs some opportunity to take up glutamate without competing with iGluSnFR. This could be accomplished by, for example, expressing iGluSnFR only in some subset of interneuron plasma membranes. To test this idea, we confined EAATs and iGluSnFRs in simulations to alternating spherical shells of varying thickness, with EAATs occupying the innermost shell in each case (*Figure 9A,B*). EAAT and iGluSnFR concentrations within each shell were adjusted so that the average concentrations were 100 and 300 µM, respectively (*Figure 9B*). As expected, excluding iGluSnFR from the region immediately surrounding the release site reduced the amplitude of the indicator signal and sped the time course of glutamate uptake compared to the case in which iGluSnFRs and EAATs were perfectly co-localized (*Figure 9C,D*). Segregation did not, however, speed the iGluSnFR signal, because the subset of glutamate that diffused into the iGluSnFR-only region tended to remain there, buffered by the surrounding indicator and contributing to a prolonged iGluSnFR signal. This effect was also evident in the glutamate clearance time course, which exhibited a second, similarly slow component. The fast component of clearance, meanwhile, was similar to that observed in the absence of iGluSnFR (*Figure 9D*, dashed line). These results suggest that, when EAATs and iGluSnFRs are strongly segregated (e.g. 2 µm alternating shells), slow iGluSnFR signals may be observed even under conditions when most synaptically released glutamate is taken up quickly (red traces in *Figure 9C,D*). Similar results were obtained in simulations employing rectilinear coordinates with iGluSnFR and EAAT expression segregated alternately or randomly into 2 × 2 × 2 µm³ cubes (data not shown). These scenarios do not appear to reflect the experimental conditions reported above, however: iGluSnFR expression in astrocytic membranes slowed STCs significantly (*Figure 4*), whereas sharp segregation (2 µm shells or cubes) simulations produced STC waveforms similar in time course to those simulated in the absence of iGluSnFR (*Figure 9E*).

Finally, we tested a different scenario in which EAATs were expressed evenly in all shells and iGluSnFRs were expressed only in alternating shells (*Figure 9F,G*). This scenario may present a case in which astroglial EAATs sample extracellular space evenly but iGluSnFR is expressed in only a subset of neurons. In this case, iGluSnFR signals were larger and faster than in the segregated case, and clearance was also faster (*Figure 9C,D and H,I*). The iGluSnFR signals were larger because the same EAAT concentration, evenly distributed, presented half the EAATs in the initial shell, enabling more glutamate to reach the iGluSnFR-containing shells. iGluSnFR signals and glutamate clearance were faster because glutamate no longer became 'trapped' in iGluSnFR-only regions. Nonetheless, the simulated STC was very similar to that in the absence of iGluSnFR (*Figure 9J*). Similar results were obtained using Cartesian coordinates (data not shown). These results indicate that iGluSnFR, when expressed in only a subset of neuronal membranes, may produce detectable (albeit smaller) signals with reduced distortion of endogenous glutamate uptake dynamics. Such an arrangement may be ideal for experiments in which many synapses can be imaged simultaneously but glutamate diffusion and uptake must not be disturbed.

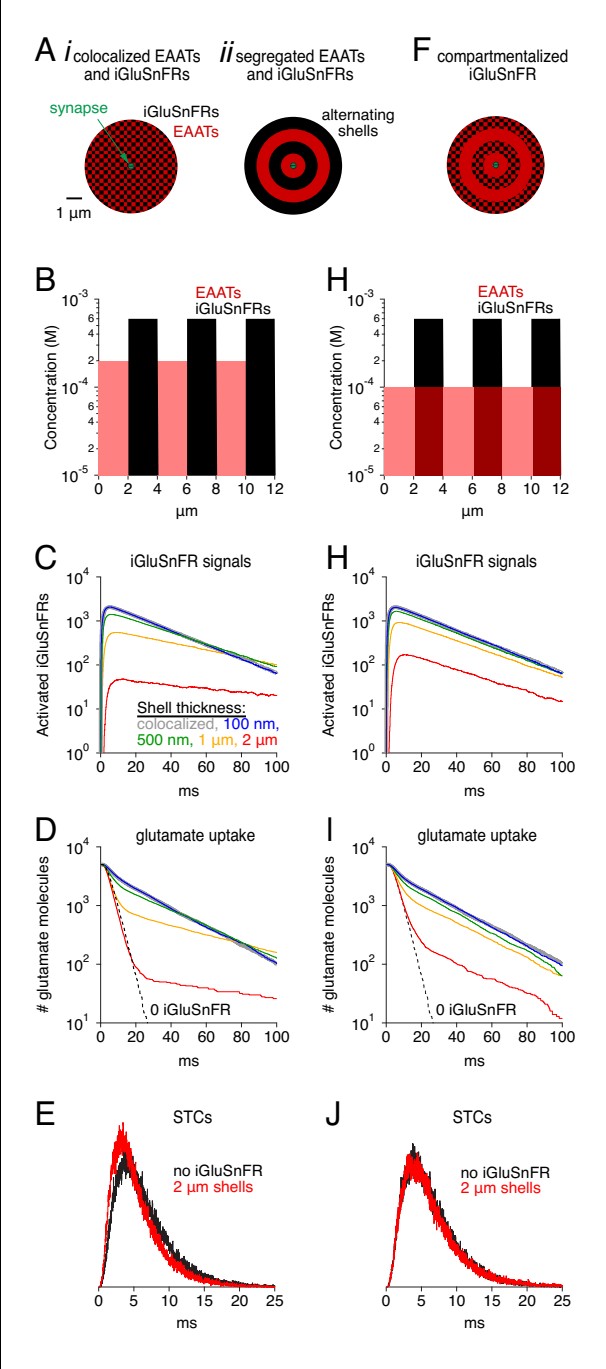

**Figure 9.** Simulations: Segregated expression reduces buffering effects. (**A**) Schematics of simulation in which EAATs and iGluSnFRs were colocalized (i), or segregated into alternating spherical shells surrounding the synapse, with EAATs occupying the innermost shell (ii). (**B**) Concentration profile of EAATs and iGluSnFRs in the 2 μm shell case. In each case, the average EAAT and iGluSnFR concentrations were 100 μM and 300 μM, respectively. (**C**) Simulated iGluSnFR signal wave forms at four different shell thicknesses. (**D**) Glutamate clearance time courses in the same simulations described in C. (**E**) Simulated synaptic transporter currents (STCs) in the 2 μm shell condition and in the absence of iGluSnFR. (**F-J**) As in *Aii-E*, except that EAATs were distributed evenly throughout all shells.

## Discussion

iGluSnFR and its variants offer great opportunities to study the dynamics of excitatory synaptic transmission in the brain. The diffusion simulations presented here aimed to examine what factors

influence the measured time course of iGluSnFR signals. They suggest that iGluSnFR responses are slower than predicted from diffusion laws and astrocyte STC recordings because iGluSnFR buffers glutamate diffusion and prolongs its extracellular lifetime following synaptic release. Consistent with these conclusions, STCs recorded from iGluSnFR$^+$ cortical astrocytes were slower than those recorded in tdTomato$^+$ control astrocytes, confirming that iGluSnFR slows glutamate clearance. The buffering effects of iGluSnFR were also evident in simulations incorporating lower affinity iGluSnFR variants. Compartmentalized iGluSnFR expression, perhaps in a subset of neuronal membranes, might reduce the buffering effect but likely would not produce faster signals that more closely approximated the native glutamate clearance time course. Despite these caveats regarding time course, multi-synapse simulations suggest that glutamate indicators provide a linear readout of the number of activated synapses.

## Indicator expression levels influence glutamate uptake and iGluSnFR signal time courses

Our simulations and experiments clearly indicate that the buffering effects and slowed glutamate clearance depend strongly on the expression levels of iGluSnFR (*Figures 2* and *4*), potentially complicating comparison of iGluSnFR signals across brain regions. For example, a recent report indicates that iGluSnFR signals are faster in the hippocampus than the cortex, suggesting that glutamate uptake in the hippocampus is more efficient (*Pinky et al., 2018*). Absent other considerations, these differences could reflect differences in iGluSnFR expression rather than uptake capacity, that is it may be that iGluSnFR is simply expressed more densely in the cortex, thereby slowing iGluSnFR signals recorded there (*Figure 2C*). In this particular case, however, STCs are also faster in hippocampus than cortex (*Hanson et al., 2015*), providing a second, complementary test that corroborates the first. Importantly, these STC recordings were made from astrocytes in the *absence* of iGluSnFR expression (*Hanson et al., 2015*). Our simulations and experiments indicate that STCs recorded in iGluSnFR$^+$ tissue also are slowed relative to control, potentially providing a misleading agreement between STCs and iGluSnFR fluorescence signals. These results also suggest that iGlu variants optimized for increased expression (*Marvin et al., 2018*), actually may disrupt glutamatergic signaling even more.

iGluSnFR concentration is likely the most critical parameter in our simulations that is not constrained in some way by experimental data. The actual effective concentration likely varies widely due to differences in brain region, expression system, promotor and indicator subtype. Biochemical measures may overestimate this parameter by including protein that is not expressed in the plasma membrane, and immunohistochemistry cannot distinguish what fraction of molecules on the cell surface is active. In the case of EAATs, quantitative immunoblotting revealed extremely high endogenous expression of EAAT1 and EAAT2 in the hippocampus and cerebellum (*Lehre and Danbolt, 1998*), and postembedding immunoelectron microscopy indicated that most EAATs are localized to astrocytic plasma membranes (*Chaudhry et al., 1995*). EAAT expression is particularly dense in the hippocampus (*Lehre et al., 1995*), exceeding 10,000 monomers per µm$^2$ of astroglial plasma membrane (*Lehre and Danbolt, 1998*). It is, admittedly, remarkable that iGluSnFR might be expressed at even higher levels, as predicted by our simulations. It is also unknown whether iGluSnFR expression in astrocytes might come at the expense of surface EAAT expression, although cortical astrocyte STCs recorded from tissue in which iGluSnFR is expressed in neurons are similar in waveform to those recorded here in iGluSnFR-expressing astrocytes (*Figure 4*; M. Armbruster and C.G. Dulla, unpublished observations).

## Effects on extrasynaptic signaling

Our simulations predicted that iGluSnFR reduces the free concentration of extrasynaptic glutamate (*Figure 2G,H*) and may therefore influence the actions of synaptically released glutamate on extrasynaptic metabotropic receptors, or perhaps glutamate spillover between excitatory synapses (*Arnth-Jensen et al., 2002*; *Scimemi et al., 2004*), thereby influencing critical aspects of synaptic signaling and plasticity.

Extrasynaptic buffering may play a particularly significant role if the target neurotransmitter typically acts at receptors located some distance from its release site. For example, dense expression of indicators for dopamine (*Sun et al., 2018*), norepinephrine, or serotonin might influence

substantially the modulatory effects of those transmitters, although dopamine has been shown to act, at least in some cases, more locally than previously expected (*Courtney and Ford, 2014*).

## Model limitations

The Monte Carlo diffusion model used here includes significant simplifications that dramatically reduce the computational resources required but may also compromise the accuracy of the results. The choice to model the extrasynaptic space as an isotropic region (*Rusakov and Kullmann, 1998*; *Diamond, 2005*), rather than instantiating an explicit structure (*Mishchenko et al., 2010*), allows the model of a single synapse to represent the average arrangement across many synapses. This approach, however, likely underestimates interactions between synapses separated by diffusion routes that are less tortuous than average. It may be that many neighboring synapses operate independently, whereas others separated by more direct diffusion routes may interact via glutamate spillover (*Arnth-Jensen et al., 2002*; *Scimemi et al., 2004*).

EAATs and iGluSnFRs were evenly distributed within spherical or cubic partitions surrounding the synapse, regular arrangements that surely differ from the endogenous structure. Similar results were obtained when partition thickness was varied from zero (i.e. continuous) to 500 nm (spherical example shown in *Figure 9*), suggesting that abstracting the fine structure of the extracellular space did not influence the results significantly. The spherical ROIs used here likely overestimates background fluorescence ($F_0$) in experiments using line scans across individual synapses (*Helassa et al., 2018*). $\Delta F/F_0$ values were considered here primarily to compare different simulation parameters, as experimentally observed $\Delta F/F_0$ values (and indicator affinity) are likely to vary with experimental and imaging conditions (*Marvin et al., 2013*; *Helassa et al., 2018*; *Marvin et al., 2018*).

Explicitly modeling only those EAATs and iGluSnFRs that bind synaptically released glutamate drastically reduced the computational power required to simulate very high levels of EAAT/iGluSnFR expression over large volumes (*Diamond, 2005*). For example, explicitly simulating 3 mM iGluSnFR in a $30 \times 30 \times 30\ \mu m^3$ volume would require tracking almost $1.5 \times 10^{11}$ Markov states (i.e. ~150 GB of RAM for iGluSnFR alone), as opposed to a maximum here of $2.2 \times 10^5$ (simulating 15 synapses releasing a total of 75,000 glutamate molecules; *Figure 8*). It did reduce binding interactions to simple probabilities (Materials and methods) at the expense of greater detail in more computationally extensive simulations (*Stiles and Bartol, 2001*) and would, therefore be insufficient to simulate steric ligand-receptor interactions or competition between individual, adjacent receptors.

## Is there an ideal glutamate indicator?

Efforts to optimize glutamate indicators generally aim to make them faster, brighter, or express more strongly (*Helassa et al., 2018*; *Marvin et al., 2018*). Our simulations suggest that increasing expression could further disrupt glutamate diffusion (*Figure 2C*) and, due to increased background fluorescence, may actually decrease $\Delta F/F_0$ (*Figure 5*). To examine what kinetic properties would yield the best performance, we combined the most advantageous kinetic properties of iGlus into one hypothetical indicator (FrankenSnFR; *Figure 10A*). A fast unbinding rate ($k_{-1}$) ensures rapid deactivation but necessitates a fast binding rate ($k_{+1}$) to maintain suitably high affinity and rapid responses at low expression levels. Both the entry and exit from the activated state ($k_{+2}$ and $k_{-2}$, respectively) must be fast to preserve rapid signal onset and cessation, and $k_{+2}$ must be significantly greater than $k_{-2}$ to achieve a high maximal activation probability ($P_{max} = k_{+2}/[k_{+2} + k_{-2}]$). Combining these features created an indicator that activated and deactivated rapidly, bound glutamate with high affinity ($k_d = 71\ \mu M$, $EC_{50} = 15\ \mu M$) and exhibited high $P_{max}$ (0.77; *Figure 10B–D*). FrankenSnFR exhibited reasonable single synapse activation at low expression levels (e.g. 3 µM, *Figure 10E*) without disrupting glutamate clearance (*Figure 10F*). Similar results were observed with $\leq 10\ \mu M$ iGluSnFR (*Figure 2*), but FrankenSnFR delivered much faster response kinetics (*Figure 10E*). Nonetheless, at higher expression levels FrankenSnFR disrupted diffusion and uptake just like other indicators. These simulations suggest that the greatest improvements in iGlu performance are gained through increasing $P_{max}$ and, of course, dynamic range ($\Delta F/F_0$), so that sufficient signal-to-noise characteristics can be achieved at low expression levels.

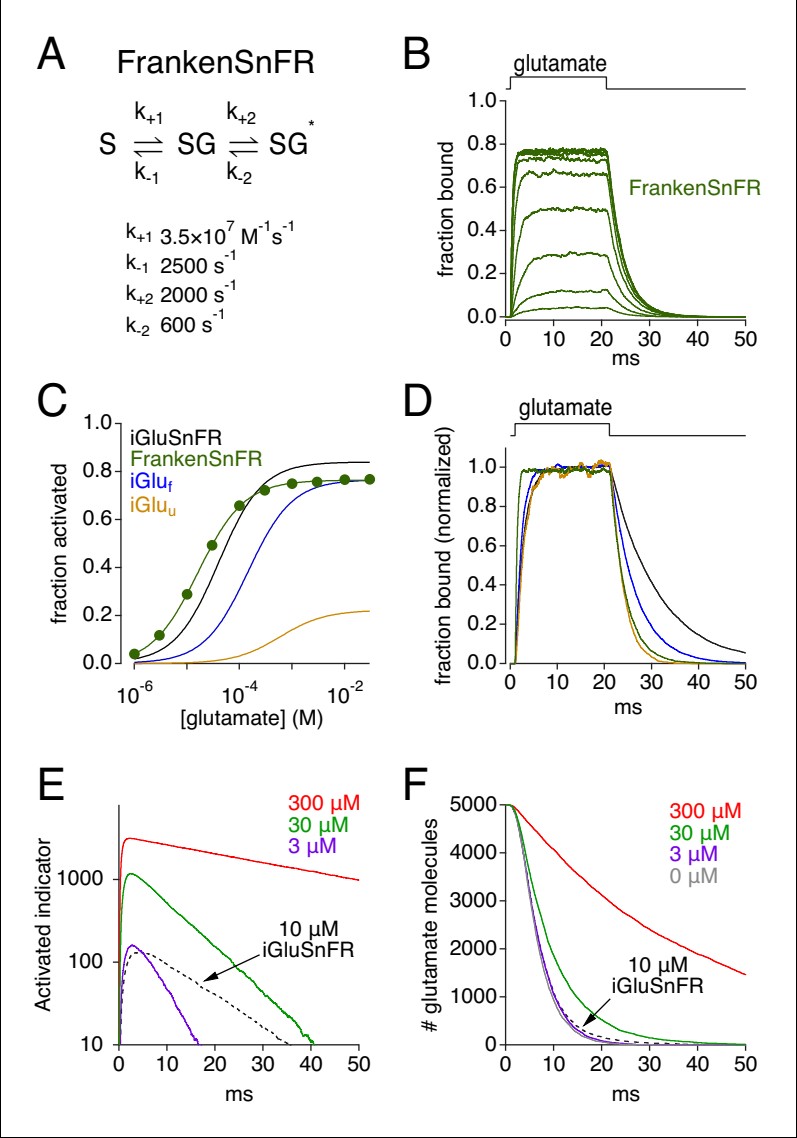

**Figure 10.** Simulations: characteristics of a theoretically ideal glutamate indicator. (**A**) Rate constants used to create FrankenSnFR, a hypothetical glutamate indicator. (**B**) Simulated FrankenSnFR activation by 20 ms applications of glutamate (concentrations varied logarithmically from 1 µM to 30 mM). (**C**) Comparison of simulated glutamate dose-response curves for iGluSnFR (black), iGlu$_f$ (blue), iGlu$_u$ (gold) and FrankenSnFR (green). (**D**) Responses of indicators (same color scheme as C) to 1 mM glutamate, normalized and superimposed to compare activation and deactivation kinetics. (**E**) Simulated FrankenSnFR responses (three different concentrations, spherical ROI radius = 10 µm) to the synaptic release of 5000 glutamate molecules. Response of 10 µM iGluSnFR shown for comparison (dashed black trace). (**F**) Glutamate uptake time course in the simulations shown in *E*, as well as clearance in the absence of any indicator (gray).

The online version of this article includes the following source data for figure 10:

**Source data 1.** | FrankenSnFR simulations.

# Materials and methods

**Key resources table**

| Reagent type (species) or resource | Designation | Source or reference | Identifiers | Additional information |
| --- | --- | --- | --- | --- |
| *Continued on next page* | | | | |

*Continued*

| Reagent type (species) or resource | Designation | Source or reference | Identifiers | Additional information |
|---|---|---|---|---|
| Strain, strain background *Mus musculus* | C57Bl/6J | Jackson Labs or in-house colony | Stock: 000664 | |
| Recombinant DNA reagent | AAV5-GFAP-iGluSnFr | Addgene/University of Pennsylvania Vector Core | Addgene: 98930-AAV5 Penn: AV-5-PV2723 | |
| Recombinant DNA reagent | AAV5-GfaABC1D-tdtomato | Addgene/University of Pennsylvania Vector Core | Addgene: 44332-AAV5 Penn: AV-5-PV3106 | |
| Recombinant DNA reagent | AAV1-hSyn-EGFP | Addgene | Cat# 50465-AAV1 | |
| Recombinant DNA reagent | AAV1-hSyn-iGluSnFr | Addgene | Cat# 98929-AAV1 | |
| Chemical compound, drug | Sulforhodamine 101 (SR-101) | Sigma | Cat# S7635-50MG | Aqueous Stock: 0.5 mM Working: 0.5 μM |
| Chemical compound, drug | DNQX | Tocris | Cat# 189 | DMSO Stock: 20 mM Working: 20 μM |
| Chemical compound, drug | AP5 | Abcam | Cat# ab120003 | Aqueous Stock: 50 mM Working: 50 μM |
| Chemical compound, drug | Gabazine/SR95531 | Tocris | Cat# 1262 | Aqueous Stock:10 mM Working: 10 μM |

All animal protocols were approved by the Tufts Institutional Animal Care and Use Committee (protocol #B2019-48).

## Adeno-associated virus injection

C57BL/6 male and female mice (P30-35) were stereotaxtically injected with 1) either GFAP-iGluSnFR or GfaABC1D-tdtomato (University of Pennsylvania Vector Core; catalog #AV-5-PV2723, AV-5-PV3106), or 2) coinfection of GFAP-iGluSnFR and hSyn-iGluSnFr (Addgene #98929-AAV1) or coinfection of GfaABC1D-tdtomato and hSyn-EGFP (Addgene #50465-AAV1) in a single hemisphere with three injections sites (coordinates): (1.25, 1.25, 0.5), (1.25, 2.25, 0.5), and (1.25, 3.25, 0.5) ($\lambda + x$, $+y$, $-z$) mm. Mice were anesthetized with isoflurane for surgery, reporter viruses were injected (1 μL per site (1:1 dilution with saline for single infection or 1:1 mix of viruses for co-infection, 0.15 μL/min) with ~5 × $10^9$ gene copies per virus. Mice were housed in 12/12 light/dark cycles following surgeries and were used for acute slice preparations 21–28 d following injection.

## Preparation of acute brain slices

Cortical brain slices were prepared from control or iGluSnFR-infected C57/B6 mice (*Armbruster et al., 2016*). Mice were anesthetized with isoflurane, decapitated, and the brains were rapidly removed and placed in ice-cold slicing solution containing (in mM): 2.5 KCl, 1.25 NaH$_2$PO$_4$, 10 MgSO$_4$, 0.5 CaCl$_2$, 11 glucose, 234 sucrose, and 26 NaHCO$_3$ and equilibrated with 95% O$_2$:5% CO$_2$. The brain was glued to a Vibratome VT1200S (Leica Microsystems, Wetzlar, Germany), and slices (400 μm thick) were cut in a coronal orientation. Slices were then placed into a recovery chamber containing aCSF comprising (in mM): 126 NaCl, 2.5 KCl, 1.25 NaH$_2$PO$_4$, 1 MgSO$_4$, 2 CaCl$_2$, 10 glucose, and 26 NaHCO$_3$ (equilibrated with 95% O$_2$:5% CO$_2$). Slices were allowed to equilibrate in aCSF at 32°C for 1 hr. Slices were loaded with sulforhodamine 101 (SR-101, 0.5 μM) in aCSF for 5 min at 32°C before equilibration (*Nimmerjahn et al., 2004*) and were allowed to return to room temperature prior to electrophysiology/imaging.

## Glutamate transporter currents

Glutamate transporter currents were recorded similarly to previous studies (*Diamond, 2005*; *Armbruster et al., 2016*). Acute slices were placed into a submersion chamber (Warner Instruments, Hamden, CT), held in place with small gold wires, and perfused with aCSF containing DNQX (20 µM), AP5 (50 µM) and Gabazine (SR95531, 10 µM) to block AMPA, NMDA and GABA$_A$ receptors, respectively. Additionally, aCSF contained BaCl$_2$ (200 µM) to block astrocyte K$^+$ conductances and isolate transporter currents (*Ransom and Sontheimer, 1995*; *Afzalov et al., 2013*; *Armbruster et al., 2016*). aCSF was equilibrated with 95% O$_2$:5% CO$_2$ and circulated at 2 ml/min at 34°C. A tungsten concentric bipolar stimulating electrode (FHC) was placed in the deep cortical layers, and astrocytes were patched and/or iGluSnFR imaged in layer II/III. Astrocytes were identified by morphology (small, round cell bodies), membrane properties, and SR-101/tdTomato labeling as imaged with a Cy3 filter cube (excitation 560/40 nm, emission 630/75 nm, Chroma). Astrocyte internal solution contained the following (in mM): 120 potassium gluconate, 20 HEPES, 10 EGTA, 2 MgATP, and 0.2 NaGTP. 4–12 MΩ borosilicate pipettes were used to establish whole-cell patch-clamp recordings using a Multiclamp 700B patch-clamp amplifier, sampled at 10 kHz using pClamp software (Molecular Devices, San Jose, CA). Once a whole-cell recording was established, cells were confirmed as astrocytes based on their passive membrane properties, low membrane resistance, and hyperpolarized resting membrane potential.

100 µs stimulus pulses were generated every 15 s with a stimulus isolator (ISO-Flex, A.M.P.I., Jerusalem, Israel). Stimulus intensity was set at 2 times the resolvable threshold stimulation. Simultaneous electrophysiology and iGluSnFR imaging were performed in a subset of cells. Imaging was performed using a Prime95b (Teledyne Photometrics, Tucson, AZ) camera with a 200 Hz frame rate, illuminated by a CoolLED illuminator and GFP filter cube (Chroma Technology, Bellows Falls, VT) and controlled by MicroManager (*Edelstein et al., 2014*) with a 60× water immersion objective (LUMPLANFL, Olympus, Waltham, MA) on an Olympus/Prior Openscope microscope. The imaged region was 97 × 37 µm$^2$ (530 × 200 pixels, 183 nm per pixel).

## Analysis

Appropriate sample sizes for astrocyte recording experiments were not determined a priori but were estimated based on previous experience with these recordings (*Armbruster et al., 2016*). All experimental n values reflect biological replicates.

Analysis was performed using MATLAB (The MathWorks, Natick, MA) and Origin (Originlab, Northampton, MA). For astrocyte synaptic transporter current recordings, 4–12 sweeps were averaged and normalized, and the decay of the glutamate transporter current was fit with a mono-exponential function (plus *y* offset) to quantify glutamate uptake kinetics (fitting region was 18–148 ms post-stimulus). For iGluSnFR imaging, 10 repeated runs of identical stimulation were averaged together and decays were fit with a bi-exponential function (decay + bleaching).

Absolute time courses of STCs and iGluSnFR signals were not compared between the single- and double infection data sets because the control STCs in the hSyn-EGFP/GfaABC1 D-tdTomato mice were significantly faster ($\tau_{decay}$ = 9.8 ± 2.3 ms, mean ± SD, n = 13) than those in the GfaABC1D-tdTomato mice (16.0 ± 3.6 ms, n = 15; $t(25.8)$ = 4.9, p=4.4 × 10$^{-5}$, *t*-test). The reason for this difference is unclear. It seems unlikely that expression of a second virus in neurons accelerated glutamate uptake in glia, but it is possible that some undetected difference in recording conditions caused a change in STC and iGluSnFR time courses. Another, perhaps most likely reason, may be that the double-infection experiments were performed on C57Bl/6 mice newly acquired from Jackson Labs, whereas the single infection experiments were performed on C57Bl/6 mice bred in-house. Whatever the cause, the time courses were consistent within each data set, so in *Figure 4G,H* the STC and iGluSnFR signal decays were normalized to the control STC decays in each data set.

## Simulations

Transmitter diffusion, uptake and glutamate indicator activation were simulated using an expanded version of a previous model (*Diamond, 2005*) written in MATLAB. Results were analyzed and graphed using IgorPro (WaveMetrics, Lake Oswego, OR).

Glutamate diffusion in single-synapse simulations was modeled as a random walk of 5000 independent glutamate molecules originating simultaneously from a point source in the center of a 320-

nm-diameter, 20 nm thick synaptic cleft (*Ventura and Harris, 1999*). At each time step $\Delta t = 1$ μs ($\Delta t = 0.5$ or 2 μs yielded similar results), each glutamate molecule was displaced in each spatial dimension by a distance r randomly selected from a normal distribution about zero (average $r^2 = 2D\Delta t$; *Hille, 1984*, where D (the diffusion coefficient)=0.253 $\mu s^2\ ms^{-1}$ in extracellular fluid at 25˚ C; *Longsworth, 1953*; *Nielsen et al., 2004*). Diffusion within the synaptic cleft was limited to two (*x, y*) dimensions (*Barbour and Häusser, 1997*); extrasynaptic diffusion was modeled three-dimensionally through an isotropic extrasynaptic space (extracellular volume fraction = 0.21). D in all spaces was reduced further to account for tortuosity of the extracellular space ($D^* = D/\lambda^2$; $\lambda = 1.55$; *Rusakov and Kullmann, 1998*; eliminating tortuosity within the synaptic cleft did not affect the results significantly). For multi-synapse simulations (*Figure 7*), synapses were modeled as point sources within isotropic extracellular space. Control simulations confirmed that removal of the synaptic cleft had little effect on the simulated iGluSnFR waveform for ROIs > 2 μm radius (*Figure 8—figure supplement 1*).

Transporters were modeled using a Markov representation of EAAT2 (*Bergles et al., 2002*; *Figure 1—figure supplement 1A*), with two simplifying modifications: The extracellular transporter was configured to bind $H^+$ prior before glutamate, rather than allowing either to bind first, and the $T_iNa_2 \rightleftharpoons T_oNa_2$ transition was eliminated. Once all the transported elements unbound on the intracellular side, the glutamate molecule was designated as taken up and removed from the simulation and the transporter returned to the unbound, outward facing state at a rate corresponding to physiological measured recovery rate (*Bergles et al., 2002*). Substrate concentrations other than $[glu]_o$ were assumed constant. Simulated STC waveforms (*Figure 2B*, inset, *Figure 9E,J*) reflected the stoichiometric current in the model (+one for forward transitions 1, 7 and 9,–1 for forward transition 15 in *Bergles et al., 2002*). Voltage-dependent rates were calculated at −95 mV (similar results were observed at −70 mV, data not shown). iGluSnFR kinetics were implemented according to a simple three-state model: bound, unbound, and fluorescent (*Helassa et al., 2018*; *Figure 1—figure supplement 1B*).

Extracellular space was partitioned transparently into 10-nm-think concentric spherical shells (single-synapse simulations) or 100 × 100 × 100 $nm^3$ cubes (multi-synapse simulations) so that local transporter, iGluSnFR and glutamate concentration could be determined. At each time step, the probability of binding to a transporter or iGluSnFR was determined independently for each glutamate molecule as follows: First, the EAAT2 and iGluSnFR glutamate binding rates (*Bergles et al., 2002*; *Helassa et al., 2018*) were multiplied by the time step, the glutamate concentration in the relevant shell/cube and the number of transporter/iGluSnFR molecules in the shell/cube, to give the number of transporters/iGluSnFRs bound in the time step, and then divided by the number of glutamate molecules in the shell/cube to yield the probability that a particular glutamate molecule would bind. If binding occurred (i.e. if a random number between 0 and 1 was less than the binding probability), the number of free transporter/iGluSnFR molecules in the cell was decremented. Once bound, the glutamate molecule underwent probabilistic transitions in subsequent time steps through the Markov schemes. Because transporters and iGluSnFRs were modeled explicitly only upon binding an individual glutamate molecule, the number of simulated transporters/iGluSnFRs was limited by the relatively low number (5000–75,000) of glutamate molecules simulated.

## Acknowledgements

This work was supported by the NIH (NS113499, NS104478, NS100796 to CGD) and the NINDS Intramural Research Program (NS003039 to JSD). We thank David DiGregorio for helpful comments on the manuscript. The authors have no competing interests to disclose.

## Additional information

### Funding

| Funder | Grant reference number | Author |
| --- | --- | --- |
| National Institute of Neurological Disorders and Stroke | NS003039 | Jeffrey S Diamond |

| National Institute of Neurological Disorders and Stroke | NS113499 | Chris G Dulla |
| National Institute of Neurological Disorders and Stroke | NS104478 | Chris G Dulla |
| National Institute of Neurological Disorders and Stroke | NS100796 | Chris G Dulla |

The funders had no role in study design, data collection and interpretation, or the decision to submit the work for publication.

## Author contributions

Moritz Armbruster, Conceptualization, Data curation, Formal analysis, Investigation, Methodology, Writing - review and editing; Chris G Dulla, Conceptualization, Resources, Supervision, Funding acquisition, Project administration, Writing - review and editing; Jeffrey S Diamond, Conceptualization, Resources, Data curation, Software, Formal analysis, Supervision, Funding acquisition, Validation, Investigation, Visualization, Methodology, Writing - original draft, Project administration, Writing - review and editing

## Author ORCIDs

Chris G Dulla  http://orcid.org/0000-0002-6560-6535
Jeffrey S Diamond  https://orcid.org/0000-0002-1770-2629

## Ethics

Animal experimentation: All animal protocols were approved by the Tufts Institutional Animal Care and Use Committee (protocol #B2019-48).

## Decision letter and Author response

Decision letter https://doi.org/10.7554/eLife.54441.sa1
Author response https://doi.org/10.7554/eLife.54441.sa2

# Additional files

## Supplementary files

• Source code 1. Matlab code for radially symmetric diffusion simulations of release from a single synapse. Simulated random walk diffusion of neurotransmitter molecules from the center of a synaptic cleft surrounded by radially symmetric, isotropic three-dimensional extrasynaptic space. User can vary cleft dimensions, the number of transmitter molecules released, iGluSnFR subtype and concentration, EAAT concentration, etc. Provides graphical updates of simulation progress and writes output files. Allows multiple trials to be averaged together.

• Source code 2. Matlab code for radially symmetric diffusion simulations of release fom multiple synapses. Simulated random walk diffusion of neurotransmitter molecules from a variable number of point sources located within isotropic extrasynaptic space. Space is represented in cartesian coordinates, and diffusion distance is measured from each release point. User can vary the number and location of synapses, the number of transmitter molecules released, iGluSnFR subtype and concentration, EAAT concentration, etc. Provides graphical updates of simulation progress and writes output files. Allows multiple trials to be averaged together.

• Transparent reporting form

## Data availability

Source data files for simulations and physiological experiments are available via Dryad (https://doi.org/10.5061/dryad.573n5tb4q).

The following dataset was generated:

| | Database and |

| Author(s) | Year | Dataset title | Dataset URL | Identifier |
|---|---|---|---|---|
| Armbruster M, Dulla CG, Diamond JS | 2020 | Data from: Effects of fluorescent glutamate indicators on neurotransmitter diffusion and uptake | https://doi.org/10.5061/dryad.573n5tb4q | Dryad Digital Repository, 10.5061/dryad.573n5tb4q |

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
