## [Decision Letter]

**Acceptance summary:**

Modern physiology in neuroscience is increasingly accomplished using genetically encoded optical indicators. Despite the explanatory power of such tools, the potential impacts of expressing such foreign sensors on physiological function are often overlooked. In Armbruster et al., the authors present a comprehensive study using physiology and modeling to illustrate the consequences of iGluSnFR (genetically encoded fluorescent glutamate indicator) expression on glutamate handling in the synaptic cleft. In particular they find that iGluSnFr expression affects synaptically-evoked glutamate diffusion, effectively slowing it by providing an exogenous buffer.

**Decision letter after peer review:**

Thank you for submitting your article "Effects of fluorescent glutamate indicators on neurotransmitter diffusion and uptake" for consideration by *eLife*. Your article has been reviewed by two peer reviewers, and the evaluation has been overseen by a Reviewing Editor and Olga Boudker as the Senior Editor. The following individual involved in review of your submission has agreed to reveal their identity: Matthew J M Rowan.

The reviewers have discussed the reviews with one another and the Reviewing Editor has drafted this decision to help you prepare a revised submission.

Summary:

Modern neuroscience is increasingly accomplished using genetically encoded optical indicators that can provide real time readouts of cellular function that can be utilized in behaving animals. Despite the ability of these tools to accelerate scientific progress, the consequences of their introduction into a functioning organism are often overlooked. The authors here present a comprehensive overview of the consequences of iGluSnFR (genetically encoded fluorescent glutamate indicator) expression on glutamate handling in the synaptic cleft. Because iGluSnFR can expressed at very high concentrations, it represents a very significant glutamate buffer. This will slow glutamate-driven processes. Thus, both glutamate transporter currents and optical readouts will be slowed. This manuscript combines physiology and computational modeling to demonstrate this important confound that must be considered when using this and other sensor tools.

Overall, the manuscript details a number of effects that could and probably already have led to misinterpretations of glutamate-signalling processes probed with iGluSnFR. Both reviewers were convinced by the results which nicely complement experimental and theoretical approaches. The manuscript could be strengthened by additional biological experiments or additional discussion.

Essential revisions:

Potential improvements include the following:

– Varying the expression level of iGlu for comparison in cortical astrocytes would further complement the model simulations. For example, the authors could inject at a higher (or lower) viral titre initially and compare uptake kinetics at day 21-28 post surgery.

– The relative ratio of iGlu and EAATs is likely a crucial component of glutamate clearance. Based on the ephys results, can the authors predict whether there is an upregulation of EAAT expression in response to iGlu expression (i.e., what is the difference in resting membrane potential in iGlu+ astrocytes due to)? Perhaps look at RMP changes following pharmacological block of EAATs in each condition?

– How might the prolongation of excitatory neurotransmission with iGlu's affect the activation of different subsets of glutamate receptors, i.e., NMDA/AMPA during the time course in the cleft? Relatedly, what is known regarding the potential excitotoxic effects of iGlu overexpression?

---

## [Author Response]

Essential revisions:Potential improvements include the following:– Varying the expression level of iGlu for comparison in cortical astrocytes would further complement the model simulations. For example, the authors could inject at a higher (or lower) viral titre initially and compare uptake kinetics at day 21-28 post surgery.

In response to this suggestion, we generated mice in which iGluSnFR was expressed in both neurons and astrocytes, rather than astrocytes alone. We chose to increase the effective iGluSnFR expression rather than decrease it, because our iGluSnFR signals are already on the fast end of the published range using large-scale imaging methods, suggesting relatively low expression. In one group of animals, iGluSnFR was expressed under control of the hSyn (neuronal) and GFAP (primarily glial) promoters. Another control group was infected with hSynEGFP and GfaABC1D -tdTomato (GfaABC1D is a slightly abbreviated version of the GFAP promoter). As predicted by our simulations, iGluSnFr expression slowed STCs to a greater degree, compared with controls, when iGluSnFr was expressed in both neurons and astrocytes as compared to when iGluSnFr was expressed in astrocytes alone (new Figure 4).

We did not directly compare the absolute time courses of STCs or iGluSnFR signals between the single- and double infection data sets because the control STCs in the hSyn-EGFP/GfaABC1D-tdTomato mice were significantly faster (τ_decay_ = 9.8 ± 2.3 ms, mean ± SD, n=13 ) than those in the previous single-infection data set (16.0 ± 3.6 ms, n=15, *p*=4.4×10^-5^, *t*-test). In the Materials and methods, we consider three possible explanations:

1) The addition of a second virus to target neurons may have caused glutamate uptake to be accelerated in glia. We think this possibility is unlikely.

2) Although we carefully maintained similar superfusion environments in the two sets of experiments, the second set of experiments may nonetheless have been performed under some slightly different condition that gave rise to faster STC and iGluSnFR time courses.

3) To complete these experiments within the time frame prescribed by the editors, we ordered appropriately aged C57Bl/6 mice from Jackson labs for the double infection experiments, whereas the previous single-infection experiments were performed with an inhouse C57Bl/6 line. Despite being genetically similar, these new mice may have exhibit faster uptake kinetics.

We favor the third possibility but, in any case, the time courses were consistent within each data set, so we have chosen to normalize the STC and iGluSnFR signal time courses to the control STC decays in each data set.

– The relative ratio of iGlu and EAATs is likely a crucial component of glutamate clearance. Based on the ephys results, can the authors predict whether there is an upregulation of EAAT expression in response to iGlu expression (i.e., what is the difference in resting membrane potential in iGlu+ astrocytes due to)? Perhaps look at RMP changes following pharmacological block of EAATs in each condition?

The small differences in RMP are unlikely to be due to changes in EAAT expression. First, as now noted in the Results, astrocyte RMP in P21 astrocytes was not substantially different in WT (-73.6 ± 1.7 mV n=12), EAAT1^+/-^ (-74.2 ± 1.6 mV; n=17; *p*=0.96 compared to WT) and EAAT2^+/-^ (-71.4 ± 1.5 mV; n=12; *p*=0.35 compared to WT) animals (J. Shih and C. Dulla personal communication). This comparison is analogous to experiments in which transporter function is reduced pharmacologically. (For the reasons highlighted in the new Figure 6, we consider it problematic to evaluate the effects of completely blocking transporters.) In addition, astrocyte RMP was not substantially different in control astrocytes (from hSyn/EGFP/ GfaABC1DtdTomato mice) and in astrocytes from hSyn-iGluSnFR/GFAP-iGluSnFR animals (control: -72.4 ± 1.4 mV, n=18; iGluSnFr -71.5 ± 1.3 mV n=20; *p*=0.66, *t*-test). These issues are now considered in the Results along with the rest of the data in Figure 4.

– How might the prolongation of excitatory neurotransmission with iGlu's affect the activation of different subsets of glutamate receptors, i.e., NMDA/AMPA during the time course in the cleft? Relatedly, what is known regarding the potential excitotoxic effects of iGlu overexpression?

We know of no excitotoxic effects of iGluSnFR expression in the literature. New simulations suggest that iGluSnFR expression likely reduces pathological activation of glutamate receptors that could lead to excitotoxicity (Figure 3). If anything, then, iGluSnFR may be neuroprotective. There is, of course, the possibility that AAV expression causes the tissue to become unhealthy, but we controlled for this by comparing between iGluSnFR^+^ and tdTomato^+^ astrocytes.

We now include simulations suggesting that glial iGluSnFR expression does not affect the glutamate time course or receptor activation in the cleft (processes controlled predominantly by diffusion), and the rapid diffusion, dilution and clearance of glutamate predicts little AMPAR or NMDAR activation beyond the cleft in which glutamate is released (new Figure 3). These simulations do suggest, however, that glial iGluSnFR expression may reduce perisynaptic mGluR activation (new Figure 3B*v*), potentially influencing homosynaptic modulation.

In addition, we have performed simulations in which iGluSnFR is present in the synaptic cleft as well as extrasynaptic regions. We find that iGluSnFR in the cleft does not substantially affect synaptic receptor activation (Figure 3D).

The rapid glutamate dilution and limited extrasynaptic receptor activation in our simulations are consistent with previous work (Wahl et al., 1996; Barbour, 2001). We employed an experimentally verified diffusion coefficient (Nielsen et al., 2004), but we wanted to ensure that our simulation results did not depend critically on this parameter. We therefore repeated simulations with a five-fold lower diffusion coefficient (new Figure 3—figure supplement 1). As expected, slower diffusion led to slower dilution and enhanced receptor activation in the synaptic cleft and perisynaptically, but the effects of iGluSnFR on receptor activation remained the same. In addition, this change in glutamate diffusion coefficient caused only very minor changes in simulated iGluSnFR signals or uptake time course (Figure 3—figure supplement 1D, E).